# HHD-GP: Incorporating Helmholtz-Hodge Decomposition into Gaussian Processes for Learning Dynamical Systems

**Hao Xu**[1,2]**, Jia Pan**[1,2,3]*
[1]The University of Hong Kong, Hong Kong, China
[2]Centre for Transformative Garment Production, Hong Kong, China
[3]LimX Dynamics
xuhaovic@connect.hku.hk, jpan@cs.hku.hk

## Abstract

Machine learning models provide alternatives for efficiently recognizing complex patterns from data, but the main concern in applying them to modeling physical systems stems from their physics-agnostic design, leading to learning methods that lack interpretability, robustness, and data efficiency. This paper mitigates this concern by incorporating the Helmholtz-Hodge decomposition into a Gaussian process model, leading to a versatile framework that simultaneously learns the curl-free and divergence-free components of a dynamical system. Learning a predictive model in this form facilitates the exploitation of symmetry priors. In addition to improving predictive power, these priors make the model indentifiable, thus the identified features can be linked to comprehensible scientific properties of the system. We show that compared to baseline models, our model achieves better predictive performance on several benchmark dynamical systems while allowing physically meaningful decomposition of the systems from noisy and sparse data.

## 1 Introduction

A dynamical system describes how the state of a system evolves over time [64]. Data-driven modeling of dynamical systems has become a fundamental task in many modern science and engineering applications, such as physical emulation [21] and robotics control [13]. Mathematically, a dynamical system is often a set of first-order ordinary differential equations (ODEs) or, equivalently, a smooth vector field on a manifold [28]. Given the functional form of ODEs, classical data-driven methods typically involve optimizing their parameters [52]. However, for many complex systems it is practically difficult to determine the form of the ODEs governing their dynamics.

Recent advances in machine learning (ML) focus on the use of neural networks [11] and non-parametric Bayesian models [63, 27, 26] for the black-box approximation of vector fields. Despite the rich expressive power, the physics-agnostic design of these models hinders their performance when applied to physical systems. To address this issue, a popular approach is to develop ML models that incorporate strong physical priors as inductive biases. Such prior knowledge commonly stems from basic physical principles related to certain differential invariants of vector fields. For example, in scenarios of learning Hamiltonian systems [23, 67, 54] and incompressible fluid dynamics [73, 33], ML models are constructed to learn divergence-free (div-free) vector fields, as a consequence of conservation laws of energy or mass. By constraining the solution space, the powerful physical principles effectively improve the extrapolation performance of ML models and enhance their interpretability. However, enforcing the model's behavior to adhere to certain fundamental physical

---

*Corresponding author.

38th Conference on Neural Information Processing Systems (NeurIPS 2024).

principles also restricts the applicability of the model. For example, a div-free vector field fails to describe a dynamical system with dissipation, but real-world dynamical systems always suffer from non-negligible dissipation.

To develop a predictive model covering more dynamical systems, we explore supplementing the div-free vector field with a curl-free vector field. This is inspired by the Helmholtz-Hodge decomposition (HHD) [3, 44, 8], which states that any sufficiently smooth vector field can be expressed as the sum of a curl-free vector field and a div-free vector field. HHD is widely used in the study of Navier-Stokes equations [45, 20, 9], but in this work we explore its connections with more general dynamical systems. For example, as shown in Fig. 1, HHD can be used to characterize the dynamics of a dissipative Hamiltonian system, where the div-free component represents its conservative dynamics and the curl-free component describes the friction-induced dissipation. In addition to dissipative Hamiltonian systems, HHD is also widely used to study chaotic systems and ocean dynamics. The connection between HHD and these dynamical systems is described in more detail in Appendix A.

In this work, we construct Gaussian process (GP) priors for div-free and curl-free dynamics separately, resulting in an additive GP model in the form of HHD for learning vector fields of dynamical systems. Our resulting HHD-GP allows one to leverage more prior knowledge than modeling the system as a whole. In particular, we investigate its potential in exploiting priors of symmetries, motivated by the observation that the div-free and curl-free components of a dynamical system usually exhibit more symmetry properties than the system itself.

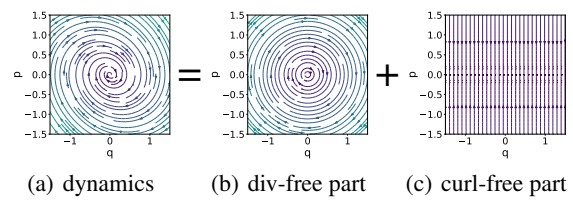

(a) dynamics     (b) div-free part     (c) curl-free part

Figure 1: The vector field in phase space of a mass-spring system and its HHD.

For example, the damped mass-spring system in Fig. 1(a) exhibits odd symmetry, but its div-free (Fig. 1(b)) and curl-free (Fig. 1(c)) components additionally present rotation and translation symmetry, respectively. Therefore, we further build symmetry-preserving div-free GPs and curl-free GPs by exploiting closure of GPs under linear transformation. The symmetry prior not only improves the predictive performance of HHD-GP, but also makes it identifiable, thus identified div-free and curl-free features can be physically-meaningful. In particular, the learned div-free features are closely related to the energy of dynamical systems.

The main contributions of this work are summarized as:

- We introduce a GP prior (called HHD-GP) for learning the Helmholtz-Hodge decomposition of a general dynamical system.

- We construct a symmetry-preserving extension of the HHD-GP that can learn physically meaningful representations of dynamical systems.

- Experiments on several benchmark systems show that our model can both accurately predict their dynamics and HHDs from sparse, noisy observations.

## 2   Related work

**Learning with div/curl-free constraints**   Div-free vector fields are a focal point of mathematical physics and have been well exploited by machine learning models to learn system dynamics, with the most well-known examples being neural networks (NNs) [23, 67] and GPs [54, 59] for learning Hamiltonian dynamics. All Hamiltonian vector fields are div-free, but not vice versa. To learn more general div-free vector fields as solutions of the continuity equation, [57] introduced an NN architecture to parameterize a universal representation of div-free vector fields. Based on the same representation, we constructed div-free kernels for GPs from matrix-valued kernels, which is an extension of the method of constructing div-free kernels from scalar kernels [50, 41, 75]. According to Maxwell's equations, div-free kernels were combined with curl-free kernels in [72, 71] to model magnetic fields. They assumed direct access to noisy observations of the div-free and curl-free components separately, whereas in this work, our goal is to recover the individual components from noisy observations of their sum. A similar prediction problem was studied in [6], they combined 2D

div-free and curl-free GP priors in the form of HHD to reconstruct planar ocean current fields, and recovered their divergence. Their model has the same formulation as ours when the dimension of HHD-GP is two, but we further developed a symmetry-preserving extension of HHD-GP to solve its non-identifiability problem. Another similar work is Dissipative Hamiltonian neural network (D-HNN) [22], which compensated HNN [23] with a curl-free part to model both conservative and dissipative dynamics simultaneously, but D-HNN is only applicable to dissipative Hamiltonian systems due to its construction.

**Learning with symmetry**   Symmetries are another important aspect of priors that can be incorporated into machine learning models. Motivated by the success of the translation-invariant NNs [38], network architectures with symmetries to more general transformations have been proposed, such as steerable CNNs [74, 10] and graph NNs [46, 61] equivariant to Euclidean symmetries. They achieved great success in improving generalization of the models. With the same motivation, kernel methods incorporating symmetries have also been developed. To make predictions invariant to transformations of inputs, [24] constructed kernels using Haar integration. And based on similar integration technique, [56] developed the Group Integration Matrix Kernel (GIM-kernel) to learn equivariant functions, which was later used by [58] to learn dynamical systems with symmetries. In this work, we constructed GP models to impose Euclidean symmetries to div/curl-free vector fields, which to the best of our knowledge has not been explored by the machine learning community.

## 3   Background

### 3.1   HHD and Problem Setup

We consider an autonomous system governed by the following ODEs:

$$\dot{\mathbf{x}}(t) := \frac{\mathrm{d}\mathbf{x}(t)}{\mathrm{d}t} = \mathbf{f}(\mathbf{x}(t)) = \mathbf{f}_{curl}(\mathbf{x}(t)) + \mathbf{f}_{div}(\mathbf{x}(t)), \qquad (1)$$

which defines a vector field by assigning a vector $\mathbf{f}(\mathbf{x}) \in \mathbb{R}^n$ to every state $\mathbf{x} \in \mathbb{R}^n$. We assume that the vector field $\mathbf{f} \in L^2(\mathbb{R}^n, \mathbb{R}^n)$ is smooth, and any such vector field can be decomposed into the sum of a curl-free vector field $\mathbf{f}_{curl} : \mathbb{R}^n \to \mathbb{R}^n$ ($\nabla \wedge \mathbf{f}_{curl} = \mathbf{0}, \forall \mathbf{x} \in \mathbb{R}^n$) and a divergence-free vector field $\mathbf{f}_{div} : \mathbb{R}^n \to \mathbb{R}^n$ ($\nabla \cdot \mathbf{f}_{div} = 0, \forall \mathbf{x} \in \mathbb{R}^n$), according to the *Helmholtz–Hodge decomposition* (HHD) [44, 8]. In this work, we are interested in learning $\mathbf{f}, \mathbf{f}_{curl}$ and $\mathbf{f}_{div}$ simultaneously from a collection of noisy observations denoted by $\mathcal{D} = \{(\mathbf{x}_i, \mathbf{y}_i)\}_{i=1}^m$, with a noisy observation $\mathbf{y}_i$ at a state $\mathbf{x}_i$ given by

$$\mathbf{y}_i = \mathbf{f}(\mathbf{x}_i) + \epsilon, \epsilon \overset{\mathrm{i.i.d}}{\sim} \mathcal{N}(\mathbf{0}, \Omega), \qquad (2)$$

where an additive noise $\epsilon \in \mathbb{R}^n$ follows a zero-mean Gaussian distribution defined by a covariance matrix $\Omega = \mathrm{diag}(\sigma_1^2, \ldots, \sigma_n^2) \in \mathbb{R}^{n \times n}$ modeling noise variance in each output dimension.

### 3.2   Vector-valued GP Model

We are interested in using Gaussian processes (GPs) to infer unknown vector fields. A GP is a stochastic process commonly used as a distribution for functions, assuming that any finite number of function values has a joint Gaussian distribution [53]. To learn an unknown vector field $\mathbf{f} : \mathbb{R}^n \to \mathbb{R}^n$, we assume a vector-valued GP prior as $\mathbf{f}(\mathbf{x}) \sim \mathcal{GP}(\mathbf{0}, \kappa(\mathbf{x}, \mathbf{x}'))$, where the mean of the function values is set to zero, and the covariance is captured by a matrix-valued kernel $\kappa : \mathbb{R}^n \times \mathbb{R}^n \to \mathbb{R}^{n \times n}$, whose $(i, j)$-th entry expresses the correlation between the $i$-th dimension of $\mathbf{f}(\mathbf{x})$ and the $j$-th dimension of $\mathbf{f}(\mathbf{x}')$. In the GP framework, the kernel controls the properties of possible functions under a GP, leading to various efforts of problem-specific design of kernels [76, 16, 18]. And to be a valid covariance function, the kernel should be symmetric and positive semidefinite [78].

GPs provide a Bayesian non-parametric approach for solving regression tasks. According to the GP prior, function values at inputs $\mathbf{X} = [\mathbf{x}_1, \ldots, \mathbf{x}_m]^\top$ are jointly distributed as $\mathcal{N}(\mathbf{f}(\mathbf{X}); \mathbf{0}, \mathbf{K})$, where $\mathbf{K} = [\kappa(\mathbf{x}_i, \mathbf{x}_j) \in \mathbb{R}^{n \times n}]_{i,j=1}^m$ is a block-partitioned covariance matrix. Then the marginal likelihood of the noisy observations $\mathbf{Y} = [\mathbf{y}_1, \ldots, \mathbf{y}_m]^\top$ given by Eq. 2 can be calculated by

$$p(\mathbf{Y} \mid \mathbf{X}) = \int p(\mathbf{Y} \mid \mathbf{X}, \mathbf{f}) p(\mathbf{f} \mid \mathbf{X}) d\mathbf{f} = \mathcal{N}(\mathbf{Y} \mid \mathbf{0}, \mathbf{K}_*), \qquad (3)$$

where $\mathbf{K}_* = \mathbf{K} + \Sigma$, $\Sigma = \Omega \otimes I_m$ is a diagonal matrix whose elements are the variance of the observation noise. Training a GP model refers to maximizing the log of Eq. 3 to optimize the kernel parameters and the noise variance. Then by conditioning on these observations using Bayes' rule, the predictive posterior for a new state $\mathbf{x}_*$ is still Gaussian with its mean $\mu$ and variance $v$ given by

$$\mu\left(\mathbf{x}_*\right) = \mathbf{k}^\mathsf{T}\mathbf{K}_*^{-1}\mathbf{Y}, \ v\left(\mathbf{x}_*\right) = \kappa\left(\mathbf{x}_*, \mathbf{x}_*\right) - \mathbf{k}^\mathsf{T}\mathbf{K}_*^{-1}\mathbf{k}, \tag{4}$$

where $\mathbf{k} = \left[\kappa\left(\mathbf{x}_1, \mathbf{x}_*\right), \ldots, \kappa\left(\mathbf{x}_m, \mathbf{x}_*\right)\right]^\mathsf{T} \in \mathbb{R}^{mn \times n}$. The derived mean function is used for regression results, and the associated variance quantifies prediction uncertainty. Due to their nonparametric nature, GPs can self-adapt to the complexity of the target function based on the data provided, without being restricted to specific parametric forms.

# 4 HHD-GP Model

We consider the problem of learning a continuous-time dynamical model in the form of HHD (Eq. 1) with GPs. HHD points out the prevalence of an additive structure in dynamical systems, so the key idea here is to exploit two GPs to model $\mathbf{f}_{curl}$ and $\mathbf{f}_{div}$, respectively, $\mathbf{f}_{curl} \sim \mathcal{GP}\left(\mathbf{0}, \kappa_{curl}\right)$, $\mathbf{f}_{div} \sim \mathcal{GP}\left(\mathbf{0}, \kappa_{div}\right)$. Then the sum of these two GPs results in a new GP modeling the dynamical system $\mathbf{f}$, with a new kernel function defined as the sum of the curl-free and divergence-free ones. And the additive kernel $\kappa = \kappa_{curl} + \kappa_{div}$ inherits the symmetric and positive-semidefinite properties of $\kappa_{curl}$ and $\kappa_{div}$, so the GP predictor (Eq. 4) is valid for the additive GP model. And the additivity of the kernels implies the additivity of GP means, so the mean function $\mu$ in Eq. 4 can be split into a curl-free part $\mu_{curl}$ and a divergence-free part $\mu_{div}$, and we have

$$\mu = \mu_{curl} + \mu_{div} = \mathbf{k}_{curl}^\mathsf{T}\mathbf{K}_*^{-1}\mathbf{Y} + \mathbf{k}_{div}^\mathsf{T}\mathbf{K}_*^{-1}\mathbf{Y}, \tag{5}$$

where $\mathbf{K}_* = \mathbf{K}_{curl} + \mathbf{K}_{div} + \Sigma$. It can be seen that the effects of $\mathbf{f}_{curl}$ and $\mathbf{f}_{div}$ can be treated as observation noises for each other, so their prediction variances at $\mathbf{x}_*$ are obtained by

$$v_{curl}\left(\mathbf{x}_*\right) = \kappa_{curl}\left(\mathbf{x}_*, \mathbf{x}_*\right) - \mathbf{k}_{curl}^\mathsf{T}\mathbf{K}_*^{-1}\mathbf{k}_{curl}, \ v_{div}\left(\mathbf{x}_*\right) = \kappa_{div}\left(\mathbf{x}_*, \mathbf{x}_*\right) - \mathbf{k}_{div}^\mathsf{T}\mathbf{K}_*^{-1}\mathbf{k}_{div}. \tag{6}$$

Consequently, observations of $\mathbf{f}\left(\mathbf{x}\right)$ given by Eq. 2 can be used to make predictions for its hidden components $\mathbf{f}_{curl}$ and $\mathbf{f}_{div}$. So we then construct GPs with realizations in the space of curl-free and div-free vector fields.

In the following parts, we construct the kernels $\kappa_{curl}$ and $\kappa_{div}$ from the representations of $\mathbf{f}_{curl}$ and $\mathbf{f}_{div}$ respectively, exploiting closure of GPs under linear transformation. Let $R = \mathbb{R}\left[\partial_{x_1}, \ldots, \partial_{x_n}\right]$ be the polynomial ring in the partial derivatives, and $\mathcal{L} \in R^{a \times b}$ be a matrix of differential operators acting on functions $g : \mathbb{R}^n \to \mathbb{R}^b$ distributed as $\mathcal{GP}\left(\mu\left(\mathbf{x}\right), \kappa\left(\mathbf{x}, \mathbf{x}'\right)\right)$. Then, the transformation of $g$ under $\mathcal{L}$ is again distributed as a GP with

$$\mathcal{L}g \sim \mathcal{GP}\left(\mathcal{L}\mu\left(\mathbf{x}\right), \mathcal{L}_\mathbf{x}\kappa\left(\mathbf{x}, \mathbf{x}'\right)\mathcal{L}_{\mathbf{x}'}^\mathsf{T} : \mathbb{R}^n \times \mathbb{R}^n \to \mathbb{R}^{a \times a}\right), \tag{7}$$

where $\mathcal{L}_\mathbf{x}$ and $\mathcal{L}_{\mathbf{x}'}$ denote the operation of $\mathcal{L}$ on the first and second argument of $\kappa\left(\mathbf{x}, \mathbf{x}'\right)$, respectively [29, 1, 36, 7]. To make $\mathcal{L}_\mathbf{x}\kappa\left(\mathbf{x}, \mathbf{x}'\right)\mathcal{L}_{\mathbf{x}'}^\mathsf{T}$ a valid covariance function, its underlying kernel $\kappa\left(\mathbf{x}, \mathbf{x}'\right)$ must be twice differentiable in $\mathbb{R}^n$. See Appendix B for details on linear operations on GPs.

## 4.1 Curl-free Kernel

The gradient operator ($\nabla := \left[\partial_{x_1}, \ldots, \partial_{x_n}\right]^\mathsf{T} \in R^{n \times 1}$) defines a surjective mapping from the space of smooth scalar fields to the space of curl-free vector fields [14], so $\mathbf{f}_{curl}$ can be represented by $\mathbf{f}_{curl} = \nabla V$, where $V \in C^\infty\left(\mathbb{R}^n, \mathbb{R}\right)$ is called the scalar potential of $\mathbf{f}_{curl}$. Since the gradient operation defines a linear transformation, if a GP with a scalar kernel $\kappa_V$ is assumed on $V$, the distribution of $\mathbf{f}_{curl}$ is again a GP. According to Eq. 7, the curl-free GP over $\mathbf{f}_{curl}$ is given by

$$\mathbf{f}_{curl} \sim \mathcal{GP}\left(\mathbf{0}, \kappa_{curl} = \nabla_\mathbf{x}\kappa_V\left(\mathbf{x}, \mathbf{x}'\right)\nabla_{\mathbf{x}'}^\mathsf{T}\right), \tag{8}$$

where $\kappa_{curl} : \mathbb{R}^n \times \mathbb{R}^n \to \mathbb{R}^{n \times n}$ is a matrix-valued kernel constructed by the Hessian of the scalar kernel $\kappa_V$, consisting of all second-order partial derivatives in $\mathbf{x}$ and $\mathbf{x}'$, with the entry in the $i$-th row and $j$-th column given by

$$\left[\kappa_{curl}\left(\mathbf{x}, \mathbf{x}'\right)\right]_{i,j} = \mathrm{Cov}\left[\partial V\left(\mathbf{x}\right)/\partial x_i, \partial V\left(\mathbf{x}'\right)/\partial x_j'\right] = \partial^2\kappa_V\left(\mathbf{x}, \mathbf{x}'\right)/\partial x_i\partial x_j'.$$

By this construction, if $\kappa_V$ induces a GP with realizations dense in $C^\infty\left(\mathbb{R}^n, \mathbb{R}\right)$, the set of realizations of $\mathcal{GP}\left(\mathbf{0}, \kappa_{curl}\right)$ is dense in the space of curl-free vector fields, because a surjective mapping maps dense sets to dense sets.

## 4.2 Divergence-free Kernel

A div-free vector field can be constructed from a skew-symmetric matrix field [5, 32, 57]. Specifically, let $\mathbf{A} : \mathbb{R}^n \to \mathbb{R}^{n \times n}$ be a skew-symmetric matrix-valued function, then a div-free vector field $\mathbf{f}_{div}$ can be represented by taking row-wise divergence of $\mathbf{A}$, *i.e.*,

$$\mathbf{f}_{div} = [\nabla \cdot \mathbf{A}_1, \dots, \nabla \cdot \mathbf{A}_n]^\top, \tag{9}$$

where $\mathbf{A}_i : \mathbb{R}^n \to \mathbb{R}^n$ is the $i$-th row of $\mathbf{A}$. The skew-symmetric matrix field $\mathbf{A}$ of size $n \times n$ can be compactly represented by $m = n\,(n-1)\,/2^2$ scalar functions $u_{ij} \in C^\infty\left(\mathbb{R}^n, \mathbb{R}\right)$:

$$\mathbf{A} = \begin{bmatrix} 0 & u_{12} & \dots & u_{1n} \\ -u_{12} & 0 & \dots & u_{2n} \\ \vdots & \vdots & \ddots & \vdots \\ -u_{1n} & -u_{2n} & \dots & 0 \end{bmatrix} = \sum_{i=1}^{n-1} \sum_{j=i+1}^{n} \Phi_{ij} u_{ij}, \tag{10}$$

where $\Phi_{ij} \in \mathbb{R}^{n \times n}$ is a matrix with its $(i, j)$-th entry equal to 1, $(j, i)$-th entry equal to -1, and all other entries equal to 0. Then, the div-free vector field given by Eq. 9 can be reformulated as a linear transformation:

$$\mathbf{f}_{div}\left(\mathbf{x}\right) = \sum_{i=1}^{n-1} \sum_{j=i+1}^{n} \psi_{ij} u_{ij}\left(\mathbf{x}\right) = \Psi \mathbf{u}\left(\mathbf{x}\right), \tag{11}$$

where $\psi_{ij} = \Phi_{ij} \nabla \in R^{n \times 1}$ is a column vector obtained by linearly transforming the gradient operator. The $m$ column vectors $\psi_{ij}$ are aggregated in a matrix $\Psi \in R^{n \times m}$, and the $m$ corresponding scalar functions $u_{ij}$ are collected in a vector-valued function $\mathbf{u} : \mathbb{R}^n \to \mathbb{R}^m$. $\Psi\left[\cdot\right]$ is a matrix of linear differential operators, so to use a GP to model $\mathbf{f}_{div}$, we can proceed by assuming a GP prior over $\mathbf{u} \sim \mathcal{GP}\left(\mathbf{0}, \kappa_{\mathbf{u}} : \mathbb{R}^n \times \mathbb{R}^n \to \mathbb{R}^{m \times m}\right)$, then based on the closure of GPs under linear transformation (Eq. 7), the GP prior over $\mathbf{f}_{div}$ (Eq. 11) is constructed by

$$\mathbf{f}_{div} \sim \mathcal{GP}\left(\mathbf{0}, \Psi_{\mathbf{x}} \kappa_{\mathbf{u}}\left(\mathbf{x}, \mathbf{x}'\right) \Psi_{\mathbf{x}'}^\top\right), \tag{12}$$

where $\kappa_{\mathbf{u}}$ is a scalar-valued kernel for two dimensional systems ($n = 2$, $m = 1$), and is a matrix-valued kernel for $n > 2$. Notice that this div-free kernel can be treated as a generalization of the div-free kernel derived from a scalar kernel [50, 41, 75], which can be recovered by setting $\kappa_{\mathbf{u}} = \kappa \cdot I$. Therefore, it is more expressive and flexible. Theoretically, the GP model given by Eq. 12 can be used to approximate arbitrary div-free vector fields, because the representation given by Eq. 9 is shown to be maximally expressive (*i.e.* universal) in [57].

## 4.3 Identifiability and Constraints

With the curl-free and div-free kernels, our objective is to learn physically interpretable representations of a dynamical system based on the HHD-GP model. However, the HHD is always not unique due to the existence of harmonic components $\mathbf{f}_{harm}$ (vector fields satisfying both $\nabla \wedge \mathbf{f}_{harm} = \mathbf{0}$ and $\nabla \cdot \mathbf{f}_{harm} = 0$, *e.g.*, constant vector fields). For the HHD of a dynamical system with the true functional decomposition $\mathbf{f}_{curl}^*$ and $\mathbf{f}_{div}^*$,

$$\mathbf{f} = \left(\mathbf{f}_{curl}^* + \mathbf{f}_{harm}\right) + \left(\mathbf{f}_{div}^* - \mathbf{f}_{harm}\right) \tag{13}$$

is a valid HHD for arbitrary $\mathbf{f}_{harm}$, which thus makes the HHD-GP model non-identifiable, meaning that from the same training data, we may learn different decompositions giving the same predictions. This is not desirable because we expect the learned dynamical model to be interpretable: the curl-free and div-free components $\mathbf{f}_{curl}$, $\mathbf{f}_{div}$ are physically meaningful.

To mitigate the identifiability problem in additive regression models, an effective method is to impose constraints on their component models [16, 17, 47, 42]. The imposed constraints can affect the decomposition results of the additive models. Therefore, to ensure that HHD-GP can produce a scientific decomposition, we desire constraints that respect the inherent characteristics of dynamical systems. And, as another primary goal, incorporating prior knowledge of a system into a GP model can also improve its prediction accuracy and learning efficiency. Therefore, in the next section, we present how to impose symmetry-based constraints on the curl-free and div-free GP models.

---

[2]$m$ is the number of entries above the diagonal. Each off-diagonal element of the matrix corresponds to a scalar function, with elements below the main diagonal as the negatives of those above.

# 5 Symmetry Constraints

## 5.1 Equivariance and Invariance

Symmetry is a fundamental geometric property prevalent in dynamical systems in natural [39], and is usually described by the concept of equivariance and invariance:

**Definition 5.1** (Equivariance and Invariance). Let $\mathcal{G}$ be a group acting on $\mathbb{R}^n$ through a smooth map $L : \mathcal{G} \times \mathbb{R}^n \to \mathbb{R}^n$. The dynamical system $\mathbf{f} : \mathbb{R}^n \to \mathbb{R}^n$ is said to be $\mathcal{G}$-equivariant if

$$\left(\mathbf{f} \circ L_g\right)(\mathbf{x}) = \mathbf{J}_{L_g}(\mathbf{x}) \mathbf{f}(\mathbf{x}), \forall \mathbf{x} \in \mathbb{R}^n, g \in \mathcal{G}, \tag{14}$$

where $L_g(\mathbf{x}) := L(g, \mathbf{x})$, and $\mathbf{J}_{L_g}$ denotes the Jacobian matrix of $L_g$. Then, $\mathcal{G}$ is termed the symmetry group of the dynamical system. In particular, if $\mathbf{J}_{L_g}$ is the identity matrix (*i.e.*, $\mathbf{f} \circ L_g = \mathbf{f}, \forall g \in \mathcal{G}$), the dynamical system $\mathbf{f}$ is said to be $\mathcal{G}$-invariant.

From the equivariance condition (Eq. 14) of the vector field, it follows the system's trajectory commutes with the action map. For vector fields on $\mathbb{R}^n$, the symmetry group $\mathcal{G}$ is commonly a subgroup of the Euclidean group $E(n)$, which comprises all intuitive geometric transformations in $\mathbb{R}^n$ (see Appendix C for a brief introduction). The symmetry constraints refer to that we expect the learned curl-free and div-free vector fields to be $\mathcal{G}$-equivariant. With the representation of their GP models, we demonstrate that the symmetries can be enforced by designing suitable kernels.

## 5.2 Symmetry-preserving Curl-free GP

The curl-free GP (Eq. 8) is constructed by transforming another GP over a potential function, implying that we can impose constraints of symmetry on the curl-free GP by designing a suitable potential GP. Therefore, we start by exploring how to construct potential functions to obtain curl-free vector fields with the desired equivariance. As expected, the following theorem holds:

**Theorem 5.2.** *Let $\mathcal{G}$ be a Euclidean group or its subgroup, and let $V : \mathbb{R}^n \to \mathbb{R}$ be a $\mathcal{G}$-invariant scalar function. Then, the curl-free vector field $\mathbf{f}_{curl} : \mathbb{R}^n \to \mathbb{R}^n$ defined by $\mathbf{f}_{curl}(\mathbf{x}) = \nabla V(\mathbf{x})$ is $\mathcal{G}$-equivariant.*

See Appendix D.1 for the proof. Theorem 5.2 shows that a $\mathcal{G}$-invariant scalar potential $V$ can yield a $\mathcal{G}$-equivariant gradient field, indicating that if any realization $V$ of $\mathcal{GP}(0, \kappa_V)$ is constrained to be $\mathcal{G}$-invariant, then its pushforward GP over $\nabla V \sim \mathcal{GP}\left(\mathbf{0}, \nabla_{\mathbf{x}} \kappa_V \nabla_{\mathbf{x}'}^{\mathsf{T}}\right)$ can induce the space of $\mathcal{G}$-equivariant curl-free vector fields.

It is obvious that a $\mathcal{G}$-invariant scalar potential $V$ can be constructed by integrating some non-invariant function $h : \mathbb{R}^n \to \mathbb{R}$ over the symmetry group: $V = \int_{\mathcal{G}}(h \circ L_g) dg$, where the measure $dg$ is called *Haar measure*, which exists for locally compact topological groups and finite groups. Therefore, by assuming that $h$ is distributed as $h \sim \mathcal{GP}(0, \kappa_h)$, we can construct the GP prior over the $\mathcal{G}$-invariant scalar potential as $V \sim \mathcal{GP}(0, \kappa_V)$, with its kernel $\kappa_V$ given by

$$\kappa_V = \text{Cov}\left[\int_{\mathcal{G}} h(L_g(\mathbf{x})) dg, \int_{\mathcal{G}} h(L_g(\mathbf{x}')) dg\right] = \int_{\mathcal{G}} \int_{\mathcal{G}} \kappa_h(L_g(\mathbf{x}), L_{g'}(\mathbf{x}')) dg dg'. \tag{15}$$

This kernel is called the *Haar-integration kernel* [24]. While it provides a general method for constructing kernels for $\mathcal{G}$-invariant functions, the double integral can be computationally expensive. If the kernel $\kappa_h$ is invariant to any $g \in \mathcal{G}$ in the sense that $\kappa(\mathbf{x}, \mathbf{x}') = \kappa(L_g(\mathbf{x}), L_g(\mathbf{x}'))$[3], a complexity reduction of Eq. 15 by one square-root can be performed by

$$\kappa_V = \int_{\mathcal{G}} \int_{\mathcal{G}} \kappa_h\left(\mathbf{x}, L_{g^{-1}g'}\right) dg dg' = |\mathcal{G}| \int_{\mathcal{G}} \kappa_h(\mathbf{x}, L_g) dg, \tag{16}$$

where $|\mathcal{G}| = \int_{\mathcal{G}} dg$, and it denotes the cardinality of $\mathcal{G}$ when the group is finite.

---

[3]For $\mathcal{G} \subseteq E(n)$, it holds that $\|L_g(\mathbf{x}) - L_g(\mathbf{x}')\| = \|\mathbf{x} - \mathbf{x}'\|$, for all $\mathbf{x}, \mathbf{x}' \in \mathbb{R}^n$, and $g \in \mathcal{G}$. Therefore, $\kappa(\mathbf{x}, \mathbf{x}') = \kappa(L_g(\mathbf{x}), L_g(\mathbf{x}'))$ is satisfied if $\kappa$ is an isotropic kernel, *i.e.*, $\kappa(\mathbf{x}, \mathbf{x}') = \kappa(\|\mathbf{x} - \mathbf{x}'\|)$, common examples of which are the squared exponential kernel and the Matérn class of kernels (cf. chap.4 in [53]).

## 5.3 Symmetry-preserving Divergence-free GP

To incorporate the equivariance condition (Eq. 14) into realizations of the div-free GP (Eq. 12), we construct the skew-symmetric matrix field $\mathbf{A}$ from a vector-valued function. Specifically, given a smooth vector field $\mathbf{v} \in C^{\infty}(\mathbb{R}^n, \mathbb{R}^n)$, $\mathbf{A}$ is constructed by $\mathbf{A} = \mathbf{J_v} - \mathbf{J_v}^{\mathsf{T}}$, where $\mathbf{J_v}$ denotes the Jacobian of $\mathbf{v}$ with its $(i,j)$-th entry given by $\partial v_i / \partial x_j$. Then the component function $u_{ij}$ in Eq. 10 is given by $u_{ij} = \partial v_i / \partial x_j - \partial v_j / \partial x_i$. By this construction, the symmetry of the div-free vector field $\mathbf{f}_{div}$ is governed by the symmetry of its vector potential $\mathbf{v}$. In particular, a $\mathcal{G}$-equivariant $\mathbf{v}$ can produce a $\mathcal{G}$-equivariant $\mathbf{f}_{div}$, and it is formalized in the following theorem:

**Theorem 5.3.** *Let $\mathcal{G}$ be a Euclidean group or its subgroup, and let $\mathbf{v} : \mathbb{R}^n \to \mathbb{R}^n$ be a $\mathcal{G}$-equivariant vector field. Then the divergence-free vector field $\mathbf{f}_{div} = [\nabla \cdot \mathbf{A}_1, \ldots, \nabla \cdot \mathbf{A}_n]^{\mathsf{T}}$ is $\mathcal{G}$-equivariant, where $\mathbf{A}_i$ denotes the $i$-th row of the skew-symmetric matrix-valued function $\mathbf{A} = \mathbf{J_v} - \mathbf{J_v}^{\mathsf{T}}$.*

See Appendix D.2 for the proof. By this theorem, we then proceed by assuming a GP prior over the vector potential $\mathbf{v} \sim \mathcal{GP}(\mathbf{0}, \kappa_{\mathbf{v}})$, and to constrain $\mathbf{v}$ to be $\mathcal{G}$-equivariant, we build its kernel $\kappa_{\mathbf{v}} \in \mathbb{R}^{n \times n}$ in the form of the *Group Integration Matrix kernel* (GIM-kernel) [56, 55], which is constructed by:

$$\kappa_{\mathbf{v}}(\mathbf{x}, \mathbf{x}') = \int_{\mathcal{G}} \kappa\left(\mathbf{x}, L_g\left(\mathbf{x}'\right)\right) \mathbf{J}_{L_g} dg, \tag{17}$$

where $\kappa$ is some arbitrary scalar-valued kernel satisfying $\kappa(\mathbf{x}, \mathbf{x}') = \kappa(L_g(\mathbf{x}), L_g(\mathbf{x}'))$ for all $g \in \mathcal{G}$. The GIM-kernel spans a Reproducing Kernel Hilbert Space (RKHS) of functions with the desired equivariance [55]. So we can then use $\kappa_{\mathbf{v}}$ (Eq. 17) to construct the GP prior over $\mathbf{u}$ in Eq. 11, where the covariance between components $u_{ij}$ and $u_{kq}$ is given by

$$[\kappa_{\mathbf{u}}]_{ij,kq} = \mathrm{Cov}\left[u_{ij} = \frac{\partial v_i}{\partial x_j} - \frac{\partial v_j}{\partial x_i}, u_{kq} = \frac{\partial v_k}{\partial x_q} - \frac{\partial v_q}{\partial x_k}\right]$$
$$= \frac{\partial^2}{\partial x_j \partial x_q'}[\kappa_{\mathbf{v}}]_{i,k} + \frac{\partial^2}{\partial x_i \partial x_k'}[\kappa_{\mathbf{v}}]_{j,q} - \frac{\partial^2}{\partial x_j \partial x_k'}[\kappa_{\mathbf{v}}]_{i,q} - \frac{\partial^2}{\partial x_i \partial x_q'}[\kappa_{\mathbf{v}}]_{j,k}. \tag{18}$$

Finally, this matrix-valued kernel $\kappa_{\mathbf{u}}$ is transformed by Eq. 12 to construct the div-free GP, of which the realizations are guaranteed to be $\mathcal{G}$-equivariant div-free vector fields, according to Theorem 5.3.

## 6 Experiments

We evaluated the performance of our proposed method in several representative physical systems. Through the experiments, we found that our model can not only accurately capture the system dynamics, but also predict correct decompositions.

### 6.1 Learning Dissipative Hamiltonian Dynamics

We first evaluated our method on a damped mass-spring system and a damped pendulum. Their governing equations are detailed in Appendix E.1. We generated the training data $\{(\mathbf{x}, \dot{\mathbf{x}})\}$ by randomly sampling states $\mathbf{x}$ in their phase space, and each of their derivative observations $\dot{\mathbf{x}}$ is corrupted by an additive Gaussian noise with a standard deviation of $0.05$.

The models are first evaluated in terms of learning ODEs. Specifically, our evaluation focused on the accuracy of the models in predicting state derivatives, as measured by the *root mean squared error* (RMSE), and their ability to accurately predict state trajectories over time, as indicated by the *valid prediction time* (VPT). In addition to these metrics for evaluating the regression results, we further use the *mean negative log likelihood* (MNLL) to evaluate the prediction uncertainty provided by the GP models. These evaluation metrics and the generation of test data are detailed in Appendix F. We compared our models, HHD-GP and its symmetry-preserving extension, SPHHD-GP, with Dissipative Hamiltonian neural network (D-HNN) [22], GPs involving div-free kernels for learning Hamiltonian dynamics (Div-GP) [54, 59], and GPs with Group Integration Matrix Kernels (GIM-GP) [56] that can incorporate symmetries. Another baseline is GPs with independent kernels (Ind-GP), which model each dimension of a dynamical system with an independent scalar GP. See Appendix G for the implementation details of these models.

Table 1: Comparison of our models to baselines. The results are averaged over 10 independent experiments performed by resampling the training sets and model initial parameters. The RMSE and the VPT are recorded in the scale of $\times 10^{-2}$ and in the form of mean $\pm$ standard deviation. Bold font indicates best results.

| Model | Damped Mass Spring | | | Damped Pendulum | | |
|---|---|---|---|---|---|---|
| | RMSE $\downarrow$ | VPT $\uparrow$ | MNLL $\downarrow$ | RMSE $\downarrow$ | VPT $\uparrow$ | MNLL $\downarrow$ |
| D-HNN | $35.58 \pm 6.08$ | $0.67 \pm 0.13$ | N/A | $183.38 \pm 27.92$ | $0.39 \pm 0.07$ | N/A |
| Div-GP | $21.48 \pm 38.11$ | $1.02 \pm 0.26$ | $2638.08 \pm 7917.14$ | $55.21 \pm 15.93$ | $1.09 \pm 0.24$ | $-0.01 \pm 0.36$ |
| Ind-GP | $4.15 \pm 1.38$ | $3.02 \pm 0.77$ | $-2.28 \pm 0.26$ | $80.59 \pm 38.08$ | $1.85 \pm 0.54$ | $-0.98 \pm 0.27$ |
| GIM-GP | $2.80 \pm 1.96$ | $5.28 \pm 2.09$ | $-2.80 \pm 0.45$ | $26.46 \pm 13.83$ | $2.32 \pm 0.74$ | $-1.15 \pm 0.22$ |
| HHD-GP (ours) | $4.83 \pm 1.60$ | $2.92 \pm 0.90$ | $-2.20 \pm 0.29$ | $34.64 \pm 15.68$ | $1.83 \pm 0.42$ | $-0.79 \pm 0.26$ |
| SPHHD-GP (ours) | $\mathbf{2.02 \pm 1.75}$ | $\mathbf{8.52 \pm 5.11}$ | $\mathbf{-3.16 \pm 0.85}$ | $\mathbf{13.34 \pm 6.89}$ | $\mathbf{3.18 \pm 1.37}$ | $\mathbf{-1.56 \pm 0.21}$ |

The results for 20 training data are shown in Table 1, and the results for an increasing number of training data are provided in Appendix J.1. The performance of Div-GP is limited because it can only model conservative dynamics. HHD-GP improves its performance by compensating with a curl-free kernel, which offsets the strong inductive bias imposed by the div-free kernel. And the performance of HHD-GP is better than that of another HHD-based model, D-HNN, because the low data efficiency of NNs makes it hard for D-HNN to capture dynamics using noisy and sparse training data, so actually the performance of D-HNN is worse than either of the GP methods. As another model without inductive bias, Ind-GP performs similarly to HHD-GP in most cases. Then, by incorporating symmetry priors into GPs, GIM-GP performs better than the above models but not as well as SPHHD-GP, because learning in the form of HHD allows the model to exploit more implicit symmetries in the dynamical systems. SPHHD-GP performs best overall in learning ODEs. Appendix J.2 presents the plots of trajectory predictions for each system.

Another advantage of our model is that it can decompose the dynamics into its div-free and curl-free components. According to Eq. 19, the div-free component can be used to recover the system's Hamiltonian, as long as the correct form of HHD is learned by our model. To show this, we evaluated our model by another task: we predicted the Hamiltonian $\hat{H}(\mathbf{x}_t)$ (energy) along the system's trajectory $\{\mathbf{x}_t\}$. For HHD-GP and SPHHD-GP, the Hamiltonian is predicted from a joint GP prior over the Hamiltonian and its underlying dynamics (see Appendix G.7.1 for details). Fig. 2 shows the predicted energy evolution of the two systems,

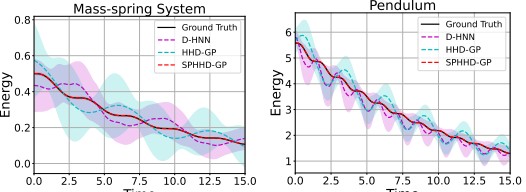

Figure 2: Energy prediction along the trajectory initialized at $(1.0, 0.0)$ of the mass-spring system (left figure) and $(1.5, 0.0)$ of the pendulum (right figure), respectively.

where we can find that D-HNN and HHD-GP fail to provide physically plausible results, because the energy should continue to decrease due to friction, but they provide oscillating predictions, along with significant variances. In contrast, predictions of SPHHD-GP are highly accurate and closely aligned with the true values. One reason is that the symmetry priors used by SPHHD-GP improves the generalization performance of the model, but more importantly, the priors solve the problem of non-identifiability suffered by HHD-GP and D-HNN. See Appendix J.3 for an visualization of predicted decompositions, which shows that although the models capture similar system dynamics, they can learn completely different decompositions.

*Consistency* is a necessary condition for a learning model to be identifiable and refers to the property that its parameter estimates should converge to the true values as the amount of data increases [19, 40]. Therefore, to further explore the non-identifiability problem, we provide the RMSE of energy prediction with an increasing number of training data in Fig. 3. The result shows that HHD-GP and D-HNN always generate significant errors in predicting energy evolution, meaning that the model parameters cannot converge as the amount of data

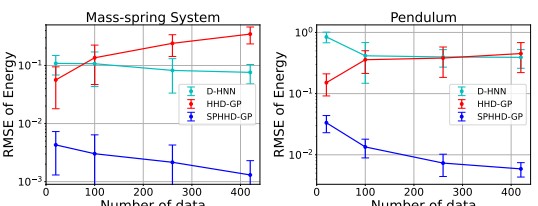

Figure 3: RMSE of energy prediction with increasing number of training data.

increases. In contrast, the energy errors of SPHHD-GP exhibits a decreasing trend, which reflects

the consistency of SPHHD-GP. In addition to this empirical validation, in Appendix H we provide a theoretical verification that for the dynamical systems in our experiments, the non-identifiability of our model (*i.e.* non-uniqueness of HHD) is solved through forced symmetries.

Accurate prediction of energy demonstrates the interpretability of learned div-free features. To further show the interpretability of curl-free features, in Appendix J.4 we added experiments of adapting the learned models to predict dynamics with unseen friction coefficients. And Appendix J.5 presents some additional experimental results of increasing the noise in training data, showing that SPHHD-GP maintains high gain in large noise compared to the baselines.

## 6.2 Learning Chaotic dynamics

Learning the correct HHD also has important implications for studying chaotic systems. [60] stated that the div-free component of a chaotic system is always orthogonal to the gradient of its energy function, *i.e.*, $\nabla H^\top \mathbf{f}_{div}(\mathbf{x}) = 0$. This energy function can be used to synchronize two chaotic systems [60], analyze their stability [77], and design energy-modulated controllers [43]. This experiment aims to learn the Chua circuit [49], which is a chaotic system with applications in various fields. Its governing equation and the HHD are detailed in Appendix E.2. The models are trained on 100 randomly sampled data, corrupted with additive Gaussian noise (standard deviation: 0.05). The generation of test data is the same as in the experiment in Section 6.1. The results are given in Table 2, which shows that SPHHD-GP consistently outperforms the other approaches. Appendix J.6 plots the trajectory and energy predictions. These results again confirm the advantage of incorporating symmetry constraints into our method. Please note that D-HNN is not applicable to this system because the div-free part of D-HNN only applies to Hamiltonian systems.

Table 2: Experimental results on the Chua circuit. $\text{RMSE}_{der}$ and $\text{RMSE}_{en}$ refer to the errors in derivative and energy predictions, respectively.

| Model | $\text{RMSE}_{der} \downarrow$ | MNLL $\downarrow$ | VPT $\uparrow$ | $\text{RMSE}_{en} \downarrow$ |
|---|---|---|---|---|
| Div-GP | $122.4 \pm 51.4$ | $0.89 \pm 0.74$ | $0.9 \pm 0.1$ | N/A |
| Ind-GP | $12.5 \pm 12.6$ | $-1.76 \pm 1.02$ | $5.9 \pm 2.2$ | N/A |
| GIM-GP | $10.2 \pm 10.6$ | $-1.92 \pm 0.67$ | $6.2 \pm 2.9$ | N/A |
| HHD-GP (ours) | $12.0 \pm 11.1$ | $-1.76 \pm 0.68$ | $5.1 \pm 1.4$ | $7.4 \pm 3.6$ |
| SPHHD-GP (ours) | $\mathbf{4.1 \pm 1.8}$ | $\mathbf{-2.72 \pm 0.29}$ | $\mathbf{13.0 \pm 5.6}$ | $\mathbf{0.7 \pm 1.5}$ |

## 6.3 Learning ocean current fields

To investigate the performance of our model in learning real-world dynamical systems, we then evaluated our model on an ocean current field, which is a complex dynamical system intricately governed by the interplay of multiple factors such as Earth's rotation, wind patterns, temperature gradients, and coastal interactions. Based on sparse observations of buoy velocities, oceanographers are interested in estimating ocean currents away from buoys and identifying divergences of the ocean current field.

In this experiment, we used a dataset from [6], containing 1183 velocity data points from 12 buoys distributed across the northern Gulf of Mexico, as shown by the red arrows in Fig. 4. We compared our model with the Helmholtz-GP model [6], which combines 2D div-free and curl-free kernels in the form of HHD to reconstruct planar ocean current fields, but without considering the severe non-identifiability problem caused by the non-uniqueness of HHD. To make the HHD of an ocean current field unique, an effective way in the field of fluid dynamics is to enforce the *parallel boundary condition*, which states that the HHD of a vector field in a bounded domain is unique if its div-free component is parallel to the domain boundary (cf. page 36 in [12], Section 5.1 in [8]). For our proposed model, the method of incorporating symmetry can be used to enforce the *parallel boundary condition*. Specifically, if a vector field has mirror symmetry with respect to a hyperplane, then the vectors on this hyperplane must be parallel to the hyperplane. Therefore, to make our model identifiable, the div-free kernel in our model was constructed to incorporate mirror symmetry with respect to the rectangular boundary of the ocean current field. To reduce the computational complexity, the kernels were fitted into a sparse GP framework [62], which reduced the computational complexity from $\mathcal{O}(m^3)$ to $\mathcal{O}(M^2 m)$, where $m = 1183$ is the number of data points and $M = 200$ is the number of inducing points. See Appendix I for the details of computational complexity of our model.

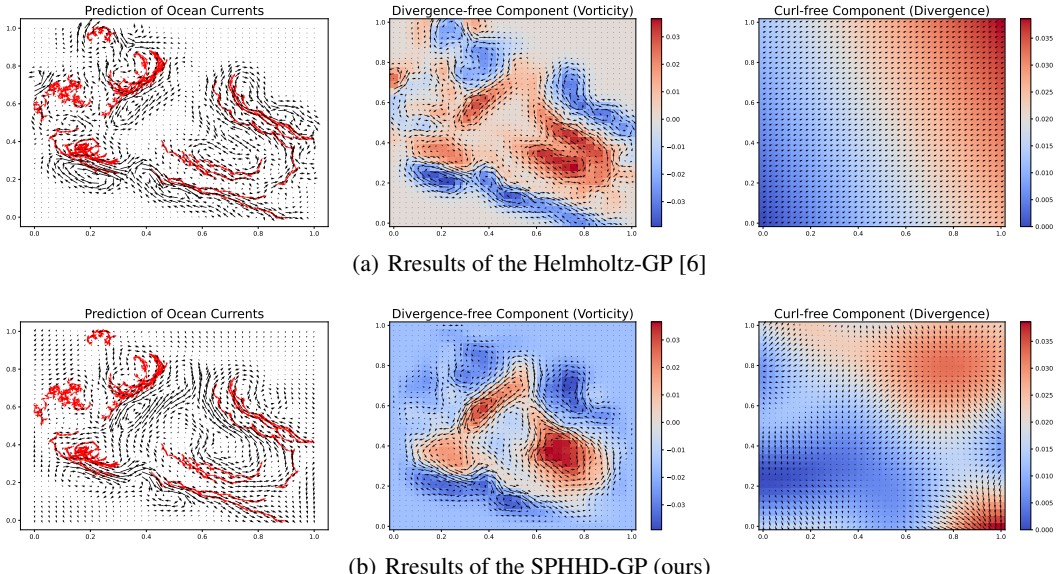

(a) Rresults of the Helmholtz-GP [6]

(b) Rresults of the SPHHD-GP (ours)

Figure 4: Predictions of the ocean current field and its HHD by (a) Helmholtz-GP and (b) SPHHD-GP. The black arrows are the predicted currents, and the red arrows denote the observed buoy velocities. First column: Predicted ocean current fields. Second column: Predicted vorticity fields. Third column: Predicted divergence fields.

The predictions of the ocean current field and its HHD are shown in Fig. 4. Although the ground truth is not available for this field of real-world ocean currents, we can still assess performance against oceanographers' expert knowledge (cf. Section I.3 in [6]). For the predictions of ocean currents (the first column of Fig. 4), the oceanographers expect to see continuous currents with no sharp deviations. However, we can clearly observe that the prediction of Helmholtz-GP shows an abrupt drop in positions away from observed data, especially in the lower left, upper right and center regions. In contrast, our model (SPHHD-GP) presents a more continuous current prediction. The predicted divergence field ($i.e.$, the scalar potential of the curl-free component) is provided in the third column of Fig. 4. The oceanographers expect to find a rich structure in divergence predictions. However, the Helmholtz-GP fails to recover the divergence field and instead predicts an almost constant curl-free vector field, as shown in the third block of Fig. 4(a). This is caused by the non-identifiability of the Helmholtz-GP, since harmonic components usually exist in the form of constant vector fields (cf. Section 4.3). Our model, instead, recovers the rich structure of the divergence field. Therefore, we can conclude that our model provides more realistic ocean current predictions and divergence identification, offering a better alternative for the simulation of ocean dynamics.

## 7   Conclusion and future work

Our work develops an additive GP model whose component is either free of divergence or of curl, the two most ubiquitous differential invariants of vector fields in natural, and we constrain the div/curl-free kernels to preserve desired symmetries. These symmetry-preserving kernels not only improve the accuracy of predictions but also make the model identifiable, thus a physically meaningful decomposition can be predicted. Our future work is to extend our model to exploit the connection of HHD with more dynamical systems. For example, there are recent advances in using the HHD to construct Lyapunov functions [65]. So, our model has potential to achieve good performance in learning stable dynamics.

## Acknowledgments

This research is partially supported by the Innovation and Technology Commission of the HKSAR Government under the InnoHK initiative and GHP/126/21GD, and GRF 17200924.

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

# Appendix

## A HHD and dynamical systems

This section provides a brief introduction to the connections between the Helmholtz-Hodge decomposition (HHD) and some dynamical systems needed to understand this work.

### A.1 HHD and dissipative Hamiltonian systems

We begin with a brief review of Hamiltonian mechanics [4]. For a dynamical system with $N$ degrees of freedom, the Hamiltonian formalism describes the system by defining a scalar function $H(\mathbf{x})$ known as the *Hamiltonian*, where the system state $\mathbf{x} = (\mathbf{q}, \mathbf{p}) \in \mathbb{R}^{2N}$ is described by generalized coordinates $\mathbf{q} \in \mathbb{R}^N$ and $\mathbf{p} \in \mathbb{R}^N$ in the phase space, corresponding to generalized position and momentum, respectively. The time evolution of $(\mathbf{q}, \mathbf{p})$ is governed by the symplectic gradient of its Hamiltonian, *i.e.*, $\dot{\mathbf{q}} = \frac{\partial H}{\partial \mathbf{p}}$, $\dot{\mathbf{p}} = -\frac{\partial H}{\partial \mathbf{q}}$. If the Hamiltonian is not explicitly time-dependent, $[\dot{\mathbf{q}}, \dot{\mathbf{p}}]^\mathsf{T}$ defines a divergence-free vector field ($\frac{\partial}{\partial \mathbf{q}} \frac{\partial H}{\partial \mathbf{p}} - \frac{\partial}{\partial \mathbf{p}} \frac{\partial H}{\partial \mathbf{q}} = 0$) whose flows conserve the Hamiltonian. By interpreting the Hamiltonian as energy, the Hamiltonian vector field can model systems with energy conservation. However, real-life systems often suffer from energy dissipation. In physical systems governed by an autonomous ODE, any energy variation occurs along a volume change in phase space and vice versa. Therefore, to account for energy dissipation, a term responsible for the volume contraction of the phase space can be introduced into the Hamiltonian dynamics. This term represents the energy lost by the system due to various dissipative forces and is typically modeled by gradient fields of some scalar functions $D(\cdot)$, so the motion equations of the dissipative Hamiltonian system can be given by

$$\mathbf{f}(\mathbf{q}, \mathbf{p}) = \underbrace{\left[\frac{\partial H}{\partial \mathbf{p}}, -\frac{\partial H}{\partial \mathbf{q}}\right]^\mathsf{T}}_{\text{divergence}-\text{free}} + \underbrace{\nabla D(\mathbf{q}, \mathbf{p})}_{\text{curl}-\text{free}}. \tag{19}$$

In a dissipative Hamiltonian system, the Hamiltonian still represents the system's total energy, but the additional damping term causes the system to lose energy over time. One common example of the damping term is induced by the *Rayleigh function*, $D(\mathbf{p}) = -\frac{1}{2}\mathbf{p}^\mathsf{T} Q \mathbf{p}$, where $Q \in \mathbb{R}^{N \times N}$ is a symmetric positive-definite matrix called the *Rayleigh dissipation matrix*. The Rayleigh function provides an elegant way to include dissipative forces—such as friction, air resistance, and viscosity—in the context of Hamiltonian mechanics. When the Hamiltonian $H(\cdot)$ represents mass instead of energy, Eq. 19 can also be used to model 2-dimensional compressible fluid dynamics, such as the dynamics of ocean current fields.

### A.2 HHD and chaotic systems

The dissipative Hamiltonian system (Eq. 19) is in an explicit form of HHD, because the Hamiltonian vector field is divergence-free and the dissipative field $\nabla D$ is curl-free. However, as its divergence-free part is governed by the Hamiltonian equations, Eq. 19 can only be used to describe a subset of even-dimensional systems. To extend its scope to study chaotics systems, [60] proposed to approach energy for dimensionless dynamical systems by using HHD as follows,

$$\begin{cases} \dot{\mathbf{x}} = \mathbf{f}(\mathbf{x}) = \mathbf{f}_{div}(\mathbf{x}) + \mathbf{f}_{curl}(\mathbf{x}), \ \mathbf{x} \in \mathbb{R}^n; \\ \nabla H^\mathsf{T} \mathbf{f}_{div}(\mathbf{x}) = 0; \ \dot{H} = \nabla H^\mathsf{T} \mathbf{f}_{curl}(\mathbf{x}), \end{cases} \tag{20}$$

where $\mathbf{f}_{div}$ and $\mathbf{f}_{curl}$ are divergence-free and curl-free components of a dynamical system, respectively. According to this criterion (Eq. 20), HHDs of chaotic systems were approached to calculate the energy for synchronizing two chaotic systems [60], to analyze the stability of chaotic systems [77], and to design energy modulation-based controllers [43].

## B GPs and linear operators

Gaussian processes (GPs) are closed under linear transformations (cf. Lemma 2.2 in [36], Lemma 2.1 in [25]). Let $\mathcal{L}$ be a linear operator acting on realizations of $g \sim \mathcal{GP}(\mu_g(\mathbf{x}), \kappa_g(\mathbf{x}, \mathbf{x}'))$, then under

the assumption that $\mathcal{L}$ commutes with expectation, the mean of $\mathcal{L}g$ is given by

$$\mathbb{E}\left[\mathcal{L}g\left(\mathbf{x}\right)\right] = \mathcal{L}\mathbb{E}\left[g\left(\mathbf{x}\right)\right] = \mathcal{L}\mu_g\left(\mathbf{x}\right), \tag{21}$$

and the covariance is such that

$$\begin{aligned}
\mathrm{cov}\left[\mathcal{L}_{\mathbf{x}}g\left(\mathbf{x}\right), \mathcal{L}_{\mathbf{x}'}g\left(\mathbf{x}'\right)\right] &= \mathbb{E}\left[\left(\mathcal{L}_{\mathbf{x}}g\left(\mathbf{x}\right) - \mathcal{L}_{\mathbf{x}}\mu\left(\mathbf{x}\right)\right)\left(\mathcal{L}_{\mathbf{x}'}g\left(\mathbf{x}'\right) - \mathcal{L}_{\mathbf{x}'}\mu\left(\mathbf{x}'\right)\right)^{\mathsf{T}}\right] \\
&= \mathcal{L}_{\mathbf{x}}\mathbb{E}\left[\left(g\left(\mathbf{x}\right) - \mu\left(\mathbf{x}\right)\right)\left(g\left(\mathbf{x}'\right) - \mu\left(\mathbf{x}'\right)\right)^{\mathsf{T}}\right]\mathcal{L}_{\mathbf{x}'}^{\mathsf{T}} \\
&= \mathcal{L}_{\mathbf{x}}\kappa_g\left(\mathbf{x}, \mathbf{x}'\right)\mathcal{L}_{\mathbf{x}'}^{\mathsf{T}}.
\end{aligned} \tag{22}$$

A common application of this technique is to construct GPs with realizations in the solution set of linear differential equations, assuming that $\mathcal{L}$ is a linear differential operator [29, 35, 36, 7, 25]. And similarly, we make use of the closure of GPs under linear differential operators to construct the curl-free and div-free kernels.

Although the transformed kernel (Eq. 22) has been widely used, its validity as a covariance function is rarely discussed in existing works. In a GP framework, a kernel $\kappa\left(\mathbf{x}, \mathbf{x}'\right) : \mathbb{R}^n \times \mathbb{R}^n \to \mathbb{R}^{m \times m}$ is a valid covariance function if it is:

*(i)* Symmetric, *i.e.*, $\kappa\left(\mathbf{x}, \mathbf{x}'\right) = \kappa\left(\mathbf{x}', \mathbf{x}\right)^{\mathsf{T}}, \forall \mathbf{x}, \mathbf{x}' \in \mathbb{R}^n$, and

*(ii)* Positive semidefinite, *i.e.*, $\sum_{ij}\mathbf{c}_i^{\mathsf{T}}\kappa\left(\mathbf{x}_i, \mathbf{x}_j\right)\mathbf{c}_j \geq 0$, for any finite set $\{\mathbf{x}_i\} \subset \mathbb{R}^n$ and $\{\mathbf{c}_i\} \subset \mathbb{R}^m$.

These two conditions for a kernel can be verified if and only if there exists a feature map $\phi\left(\mathbf{x}\right)$ such that $\kappa\left(\mathbf{x}, \mathbf{x}'\right) = \phi\left(\mathbf{x}\right)\phi\left(\mathbf{x}'\right)^{\mathsf{T}}$ [53]. So, if $\kappa_g\left(\mathbf{x}, \mathbf{x}'\right)$ is a valid kernel, it holds that

$$\mathcal{L}_{\mathbf{x}}\kappa_g\left(\mathbf{x}, \mathbf{x}'\right)\mathcal{L}_{\mathbf{x}'}^{\mathsf{T}} = \mathcal{L}_{\mathbf{x}}\phi\left(\mathbf{x}\right)\phi\left(\mathbf{x}'\right)^{\mathsf{T}}\mathcal{L}_{\mathbf{x}'}^{\mathsf{T}} = \left(\mathcal{L}_{\mathbf{x}}\phi\left(\mathbf{x}\right)\right)\left(\mathcal{L}_{\mathbf{x}'}\phi\left(\mathbf{x}'\right)\right)^{\mathsf{T}}. \tag{23}$$

Therefore, the transformed kernel $\mathcal{L}_{\mathbf{x}}\kappa_g\left(\mathbf{x}, \mathbf{x}'\right)\mathcal{L}_{\mathbf{x}'}^{\mathsf{T}}$ is guaranteed to be a valid covariance function provided that its underlying kernel $\kappa_g\left(\mathbf{x}, \mathbf{x}'\right)$ is. However, when $\mathcal{L}$ is a differential operator, $\kappa_g$ should be twice differentiable, which is satisfied by most of the standard kernels, such as the squared exponential kernel.

## C Basics for Euclidean group

This section gives the basic definitions about the *Euclidean group*. In the context of this work, the most important example of a symmetry group is the Euclidean group $E\left(n\right)$ or its subgroups. The set of all elements in $E\left(n\right)$ can be denoted as

$$E\left(n\right) = \left\{\left(A, \mathbf{b}\right) \mid A \in O\left(n\right), \mathbf{b} \in \mathbb{R}^n\right\}, \tag{24}$$

where $O\left(n\right) = \left\{A \in \mathbb{R}^{n \times n} \mid AA^{\mathsf{T}} = I\right\}$ is the *orthogonal group*. Any element $g = \left(A, \mathbf{b}\right) \in E\left(n\right)$ represents a translation followed by an orthogonal transformation, the action of $g$ on a point $\mathbf{x} \in \mathbb{R}^n$ is given by a linear mapping:

$$L_g : \mathbb{R}^n \to \mathbb{R}^n, \ \mathbf{x} \mapsto A\left(\mathbf{x} + \mathbf{b}\right). \tag{25}$$

Therefore, $E\left(n\right)$ comprises all *isometries* of a Euclidean space, *i.e.* for all $\mathbf{x}, \mathbf{x}' \in \mathbb{R}^n$ and $g \in E\left(n\right)$, we have

$$\left\|L_g\left(\mathbf{x}\right) - L_g\left(\mathbf{x}'\right)\right\| = \left\|\mathbf{x} - \mathbf{x}'\right\|, \tag{26}$$

where $\|\cdot\|$ is the Euclidean norm. All intuitive geometric transformations in $\mathbb{R}^n$ can be described by subgroups of $E\left(n\right)$, such as

1. **Translation:** The group of all translations in $\mathbb{R}^n$ is denoted by $\left(\mathbb{R}^n, +\right)$. For any $\mathbf{v} \in \mathbb{R}^n$, A translation is a transformation that moves a point $\mathbf{x} \in \mathbb{R}^n$ by $L_{\mathbf{v}}\left(\mathbf{x}\right) = \mathbf{x} + \mathbf{v}$.

2. **Rotation:** The group of all rotations in $\mathbb{R}^n$ is represented by the set of special orthogonal matrices $SO\left(n\right) = \left\{R \in O\left(n\right) \mid \det R = 1\right\}$, where a rotation matrix $R$ transforms a point by $L_R\left(\mathbf{x}\right) = R\mathbf{x}$.

3. **Reflection:** Reflections in $\mathbb{R}^n$ forms subgroups of the orthogonal group $O\left(n\right)$. Reflections correspond to mirror symmetries. They mirror points across a hyperplane. For a hyperplane with a unit normal vector $\mathbf{n}$, the action of a reflection is defined as $L_{\mathbf{n}}\left(\mathbf{x}\right) = \mathbf{x} - 2(\mathbf{x} \cdot \mathbf{n})\mathbf{n}$.

# D  Proofs

This section restates and proves the theorems in Section 5, which give the theoretical foundations that we use to enforce symmetry constraints to our HHD-GP model.

## D.1  Proof of Theorem 5.2

**Theorem 5.2.** *Let $\mathcal{G}$ be a Euclidean group or its subgroup, and let $V : \mathbb{R}^n \to \mathbb{R}$ be a $\mathcal{G}$-invariant scalar function. Then, the curl-free vector field $\mathbf{f}_{curl} : \mathbb{R}^n \to \mathbb{R}^n$ defined by $\mathbf{f}_{curl}(\mathbf{x}) = \nabla V(\mathbf{x})$ is $\mathcal{G}$-equivariant.*

*Proof.* To prove that $\nabla V(\mathbf{x})$ is $\mathcal{G}$-equivariant, by Definition 5.1 we need to show that for any $\mathbf{x} \in \mathbb{R}^n$ and $g \in \mathcal{G}$, we have

$$\nabla V \circ L_g = \mathbf{J}_{L_g} \nabla V, \tag{27}$$

where $L_g(\mathbf{x}) : \mathbb{R}^n \to \mathbb{R}^n$ is the action of $g$ on $\mathbf{x}$, and $\mathbf{J}_{L_g}$ is the Jacobian matrix of $L_g$. To show this, we first recall the chain rule of the gradient operator,

$$\nabla(V \circ L_g) = \mathbf{J}_{L_g}^{\mathsf{T}}(\nabla V \circ L_g), \tag{28}$$

where $\mathbf{J}_{L_g}^{\mathsf{T}}$ denotes the transpose of $\mathbf{J}_{L_g}$. Since $V(\mathbf{x})$ is $\mathcal{G}$-invariant, for all $g \in \mathcal{G}$ we have

$$V = V \circ L_g. \tag{29}$$

Now taking the gradient of both sides, and combining with the chain rule, we obtain

$$\nabla V = \nabla(V \circ L_g) = \mathbf{J}_{L_g}^{\mathsf{T}}(\nabla V \circ L_g). \tag{30}$$

Then by multiplying $\mathbf{J}_{L_g}$ on both sides, we have

$$\mathbf{J}_{L_g} \nabla V = \mathbf{J}_{L_g} \mathbf{J}_{L_g}^{\mathsf{T}}(\nabla V \circ L_g). \tag{31}$$

Considering $\mathcal{G}$ is a subgroup of the Euclidean group, the action of any group element $g \in \mathcal{G}$ on $\mathbb{R}^n$ can be represented by

$$L_g(\mathbf{x}) = A(\mathbf{x} + \mathbf{v}), \tag{32}$$

where $A \in O(n)$ is an orthogonal matrix. Therefore, $\mathbf{J}_{L_g} \mathbf{J}_{L_g}^{\mathsf{T}} = AA^{\mathsf{T}} = I$, where $I$ is an identity matrix, so we have $\mathbf{J}_{L_g} \nabla V = \nabla V \circ L_g$, which means that the curl-free vector field $\mathbf{f}_{curl} = \nabla V$ is $\mathcal{G}$-equivariant, as desired.

## D.2  Proof of Theorem 5.3

**Theorem 5.3.** *Let $\mathcal{G}$ be a Euclidean group or its subgroup, and let $\mathbf{v} : \mathbb{R}^n \to \mathbb{R}^n$ be a $\mathcal{G}$-equivariant vector field. Then the divergence-free vector field $\mathbf{f}_{div} = [\nabla \cdot \mathbf{A}_1, \ldots, \nabla \cdot \mathbf{A}_n]^{\mathsf{T}}$ is $\mathcal{G}$-equivariant, where $\mathbf{A}_i$ denotes the $i$-th row of the skew-symmetric matrix-valued function $\mathbf{A} = \mathbf{J}_{\mathbf{v}} - \mathbf{J}_{\mathbf{v}}^{\mathsf{T}}$.*

*Proof.* To prove this theorem, we need to show that the divergence-free vector field $\mathbf{f}_{div}$ satisfies the $\mathcal{G}$-equivariance condition, i.e., for all $g \in \mathcal{G}$ and $\mathbf{x} \in \mathbb{R}^n$,

$$\mathbf{f}_{div} \circ L_g = \mathbf{J}_{L_g} \mathbf{f}_{div}, \tag{33}$$

where $L_g(\mathbf{x}) : \mathbb{R}^n \to \mathbb{R}^n$ is the action of $g$ on $\mathbf{x}$, and $\mathbf{J}_{L_g}$ is the Jacobian matrix of $L_g$.

Define the divergence-free vector field $\mathbf{f}_{div} = [\nabla \cdot \mathbf{A}_1, ..., \nabla \cdot \mathbf{A}_n]^{\mathsf{T}}$, where $\mathbf{A}_i$ is the $i$th row of the skew-symmetric matrix field $\mathbf{A} = \mathbf{J}_{\mathbf{v}} - \mathbf{J}_{\mathbf{v}}^{\mathsf{T}}$, and $\mathbf{J}_{\mathbf{v}}$ is the Jacobian of some vector field $\mathbf{v} : \mathbb{R}^n \to \mathbb{R}^n$. Given that $\mathbf{v}$ is $\mathcal{G}$-equivariant, for all $g \in \mathcal{G}$ and $\mathbf{x} \in \mathbb{R}^n$ we have

$$\mathbf{v} \circ L_g = \mathbf{J}_{L_g} \mathbf{v}. \tag{34}$$

Now computing the Jacobian of both sides, by the chain rule of the Jacobian, we obtain

$$\mathbf{J}_{\mathbf{v} \circ L_g} = (\mathbf{J}_{\mathbf{v}} \circ L_g) \mathbf{J}_{L_g} = \mathbf{J}_{L_g} \mathbf{J}_{\mathbf{v}}. \tag{35}$$

Then applying $\mathbf{J}_{L_g}^{\mathsf{T}}$ on both sides, we have $\mathbf{J}_{L_g}^{\mathsf{T}}(\mathbf{J}_{\mathbf{v}} \circ L_g) \mathbf{J}_{L_g} = \mathbf{J}_{L_g}^{\mathsf{T}} \mathbf{J}_{L_g} \mathbf{J}_{\mathbf{v}}$. Since $\mathcal{G}$ is a subgroup of the Euclidean group, the Jacobian of the action of any group element $g \in \mathcal{G}$ on $\mathbb{R}^n$ is an orthogonal

matrix, i.e., $\mathbf{J}_{L_g} \in O(n)$. Therefore, $\mathbf{J}_{L_g}^\mathsf{T} \mathbf{J}_{L_g} = I$, where $I$ is an identity matrix, so it holds that $\mathbf{J}_\mathbf{v} = \mathbf{J}_{L_g}^\mathsf{T} (\mathbf{J}_\mathbf{v} \circ L_g) \mathbf{J}_{L_g}$. Then by substituting it into $\mathbf{A} = \mathbf{J}_\mathbf{v} - \mathbf{J}_\mathbf{v}^\mathsf{T}$, we obtain

$$\mathbf{A} = \mathbf{J}_\mathbf{v} - \mathbf{J}_\mathbf{v}^\mathsf{T} = \mathbf{J}_{L_g}^\mathsf{T} \left( \mathbf{J}_\mathbf{v} \circ L_g - \mathbf{J}_\mathbf{v}^\mathsf{T} \circ L_g \right) \mathbf{J}_{L_g} = \mathbf{J}_{L_g}^\mathsf{T} (\mathbf{A} \circ L_g) \mathbf{J}_{L_g}, \tag{36}$$

where the equation at the $(i,j)$-th entries of both sides is given by

$$\mathbf{A}_{ij} = \sum_{k=1}^n \left[ \frac{\partial (L_g)_k}{\partial x_i} \sum_{l=1}^n \frac{\partial (L_g)_l}{\partial x_j} (\mathbf{A}_{kl} \circ L_g) \right]. \tag{37}$$

Then by substituting $\mathbf{A}_{ij}$ into the construction of $\mathbf{f}_{div}$, for any $g \in \mathcal{G}$ and $\mathbf{x} \in \mathbb{R}^n$, we have

$$
\begin{aligned}
\mathbf{f}_{div} &= [\nabla \cdot \mathbf{A}_1, \quad ..., \quad \nabla \cdot \mathbf{A}_n]^\mathsf{T} \\
&= \left[ \sum_{j=1}^n \frac{\partial \mathbf{A}_{1j}}{\partial x_j}, \quad ..., \quad \sum_{j=1}^n \frac{\partial \mathbf{A}_{nj}}{\partial x_j} \right]^\mathsf{T} \\
&= \begin{bmatrix} \sum_{j=1}^n \sum_{k=1}^n \frac{\partial (L_g)_k}{\partial x_1} \sum_{l=1}^n \frac{\partial (L_g)_l}{\partial x_j} \frac{\partial (\mathbf{A}_{kl} \circ L_g)}{\partial x_j} \\ \vdots \\ \sum_{j=1}^n \sum_{k=1}^n \frac{\partial (L_g)_k}{\partial x_n} \sum_{l=1}^n \frac{\partial (L_g)_l}{\partial x_j} \frac{\partial (\mathbf{A}_{kl} \circ L_g)}{\partial x_j} \end{bmatrix} \\
&= \begin{bmatrix} \sum_{k=1}^n \frac{\partial (L_g)_k}{\partial x_1} \sum_{j=1}^n \sum_{l=1}^n \frac{\partial (L_g)_l}{\partial x_j} \frac{\partial (\mathbf{A}_{kl} \circ L_g)}{\partial x_j} \\ \vdots \\ \sum_{k=1}^n \frac{\partial (L_g)_k}{\partial x_n} \sum_{j=1}^n \sum_{l=1}^n \frac{\partial (L_g)_l}{\partial x_j} \frac{\partial (\mathbf{A}_{kl} \circ L_g)}{\partial x_j} \end{bmatrix} \\
&= \begin{bmatrix} \sum_{k=1}^n \frac{\partial (L_g)_k}{\partial x_1} \sum_{j=1}^n \sum_{l=1}^n \frac{\partial (L_g)_l}{\partial x_j} \sum_{p=1}^n \frac{\partial (L_g)_p}{\partial x_j} \left( \frac{\partial \mathbf{A}_{kl}}{\partial x_p} \circ L_g \right) \\ \vdots \\ \sum_{k=1}^n \frac{\partial (L_g)_k}{\partial x_n} \sum_{j=1}^n \sum_{l=1}^n \frac{\partial (L_g)_l}{\partial x_j} \sum_{p=1}^n \frac{\partial (L_g)_p}{\partial x_j} \left( \frac{\partial \mathbf{A}_{kl}}{\partial x_p} \circ L_g \right) \end{bmatrix} \\
&= \begin{bmatrix} \sum_{k=1}^n \frac{\partial (L_g)_k}{\partial x_1} \sum_{l=1}^n \sum_{p=1}^n \left( \frac{\partial \mathbf{A}_{kl}}{\partial x_p} \circ L_g \right) \sum_{j=1}^n \frac{\partial (L_g)_l}{\partial x_j} \frac{\partial (L_g)_p}{\partial x_j} \\ \vdots \\ \sum_{k=1}^n \frac{\partial (L_g)_k}{\partial x_n} \sum_{l=1}^n \sum_{p=1}^n \left( \frac{\partial \mathbf{A}_{kl}}{\partial x_p} \circ L_g \right) \sum_{j=1}^n \frac{\partial (L_g)_l}{\partial x_j} \frac{\partial (L_g)_p}{\partial x_j} \end{bmatrix} \\
&= \left[ \sum_{k=1}^n \frac{\partial (L_g)_k}{\partial x_1} \sum_{l=1}^n \left( \frac{\partial \mathbf{A}_{kl}}{\partial x_l} \circ L_g \right), \quad ..., \quad \sum_{k=1}^n \frac{\partial (L_g)_k}{\partial x_n} \sum_{l=1}^n \left( \frac{\partial \mathbf{A}_{kl}}{\partial x_l} \circ L_g \right) \right]^\mathsf{T} \\
&= \left[ \sum_{k=1}^n \frac{\partial (L_g)_k}{\partial x_1} (\nabla \cdot \mathbf{A}_k) \circ L_g, \quad ..., \quad \sum_{k=1}^n \frac{\partial (L_g)_k}{\partial x_n} (\nabla \cdot \mathbf{A}_k) \circ L_g \right]^\mathsf{T} \\
&= \mathbf{J}_{L_g}^\mathsf{T} [(\nabla \cdot \mathbf{A}_1) \circ L_g, \quad ..., \quad (\nabla \cdot \mathbf{A}_n) \circ L_g]^\mathsf{T} \\
&= \mathbf{J}_{L_g}^\mathsf{T} (\mathbf{f}_{div} \circ L_g).
\end{aligned} \tag{38}
$$

Then, applying $\mathbf{J}_{L_g}$ on both sides, we obtain $\mathbf{J}_{L_g} \mathbf{f}_{div} = \mathbf{f}_{div} \circ L_g$, which completes the proof.

# E  Physical systems in the experiments

This section reviews the physical systems used in our experiments, listing their governing equations, HHDs and symmetry properties.

## E.1  Dissipative Hamiltonian systems

The connection between HHD and dissipative Hamiltonian systems is reviewed in Appendix A.1. In our experiments, the models are evaluated with the following two dissipative Hamiltonian systems.

**Damped Mass Spring**  A damped mass-spring system is a mass attached to a spring that oscillates periodically around an equilibrium position. Its Hamiltonian in natural units is given by

$$H(q,p) = \frac{1}{2} \left( q^2 + p^2 \right), \tag{39}$$

where $q \in \mathbb{R}$ is its position, and $p$ is the momentum conjugate to $q$. The Hamiltonian represents the total energy of the oscillator. Without energy dissipation, the motion of the oscillator is described by the Hamiltonian vector field (div-free vector field $\mathbf{f}_{div}$) derived from the Hamiltonian:

$$\mathbf{f}_{div} := \begin{bmatrix} \dot{q} \\ \dot{p} \end{bmatrix} = \begin{bmatrix} \frac{\partial H}{\partial p} \\ -\frac{\partial H}{\partial q} \end{bmatrix} = \begin{bmatrix} p \\ -q \end{bmatrix}. \tag{40}$$

Then we show that the Hamiltonian vector field has SO(2)-equivariance. Apply a rotation by an angle $\theta$ that transforms any state vector $[q, p]^{\mathsf{T}}$ to $[q', p']^{\mathsf{T}}$, where $q' = q\cos\theta - p\sin\theta$, $p' = q\sin\theta + p\cos\theta$, then for all $\theta \in \mathbb{R}$, the equivariance condition (Eq. 14) can be easily verified by

$$\mathbf{f}_{div}(q', p') = \begin{bmatrix} p' \\ -q' \end{bmatrix} = \begin{bmatrix} \cos\theta & -\sin\theta \\ \sin\theta & \cos\theta \end{bmatrix} \begin{bmatrix} p \\ -q \end{bmatrix} = \begin{bmatrix} \cos\theta & -\sin\theta \\ \sin\theta & \cos\theta \end{bmatrix} \mathbf{f}_{div}(q, p). \tag{41}$$

Therefore, the Hamiltonian vector field is equivariant with respect to the 2D rotations. However, this rotation symmetry is broken when the system suffers from energy dissipation, $i.e.$, a dissipative field is added to the motion equations. The dissipative vector field (curl-free vector field $\mathbf{f}_{curl}$) is induced by the Rayleigh function $D(p) = \rho p^2/2$:

$$\mathbf{f}_{curl} = -\nabla_{(q,p)}D = [0, -\rho p]^{\mathsf{T}}, \tag{42}$$

where $\rho$ is the friction coefficient and we set it at 0.1. This dissipative field has the symmetries of the $q$-axis translation ($\mathbf{f}_{curl}(q+g, p) = \mathbf{f}_{curl}(q, p), \forall g \in \mathbb{R}$) and the rotation by an angle of $\pi$ radians (the so-called *odd symmetry*, $i.e.$, $\mathbf{f}_{curl}(-q, -p) = -\mathbf{f}_{curl}(q, p)$). Then by summing the curl-free vector field (Eq. 42) and the Hamiltonian vector field (Eq. 40), the dynamics of the damped harmonic oscillator is characterized by

$$\mathbf{f} = \mathbf{f}_{div} + \mathbf{f}_{curl} = [p, -q - 0.5p]^{\mathsf{T}}, \tag{43}$$

which only demonstrates the odd symmetry.

**Damped Pendulum**  A damped pendulum is a physical system consisting of a weight suspended from a pivot, subjected to a resistive force that gradually reduces its oscillation over time. By defining its state in the phase space through $[q, p]^{\mathsf{T}} \in \mathbb{R}^2$, the dynamics of a damped pendulum and its HHD can be given by

$$\mathbf{f} := \begin{bmatrix} \dot{q} \\ \dot{p} \end{bmatrix} = \begin{bmatrix} \frac{\partial H}{\partial p} \\ -\frac{\partial H}{\partial q} \end{bmatrix} - \begin{bmatrix} \frac{\partial D}{\partial q} \\ \frac{\partial D}{\partial p} \end{bmatrix} = \underbrace{\begin{bmatrix} \frac{l^2}{m}p \\ -2mgl\sin q \end{bmatrix}}_{divergence-free} + \underbrace{\begin{bmatrix} 0 \\ -\rho p \end{bmatrix}}_{curl-free} = \begin{bmatrix} \frac{l^2}{m}p \\ -2mgl\sin q - \rho p \end{bmatrix}, \tag{44}$$

where the Hamiltonian $H$ and the Rayleigh function $D$ are given by

$$H(q, p) = 2mgl(1 - \cos q) + \frac{l^2 p^2}{2m}, \ D(p) = \frac{1}{2}\rho p^2. \tag{45}$$

In our experiments, the gravitational constant $g$, the mass $m$, and the pendulum length $l$ were set as $g = 3$ and $m = l = 1$, and the friction coefficient $\rho$ was set as $\rho = 0.1$. The dynamics of the damped pendulum, as well as its div-free and curl-free components, exhibit the odd symmetry, but the curl-free part additionally exhibits the translation invariance along the $q$-axis.

### E.2  Chaotic system

The Chua circuit [49] is a simple electronic circuit that exhibits chaotic behavior, and it has applications in various fields, such as secure communication systems and random number generators. The ODE of a Chua circuit with its HHD is given by

$$\begin{bmatrix} \dot{x} \\ \dot{y} \\ \dot{z} \end{bmatrix} = \begin{bmatrix} \alpha(y - x^3 - cx) \\ x - y + z \\ -\beta y \end{bmatrix} = \underbrace{\begin{bmatrix} \alpha y \\ x + z \\ -\beta y \end{bmatrix}}_{divergence-free} + \underbrace{\begin{bmatrix} \alpha(-x^3 - cx) \\ -y \\ 0 \end{bmatrix}}_{curl-free}, \tag{46}$$

where $x$, $y$, $z$ are the phase space variables, and $\alpha$, $c$, $\beta$ are the system parameters, which were set to 10, $-0.143$, and 16 respectively in the experiments. The Chua circuit is equivariant under a $\pi$

rotation about the origin: $L_\pi : (x, y, z) \mapsto (-x, -y, -z)$. If the Chua circuit is decomposed by the HHD in Eq. 46, more symmetries are exhibited. Specifically, its div-free component inherits the odd symmetry, but additionally presents the translation invariance along $x = z$. The curl-free vector field is invariant along the $z$-axis and equivariant under mirror reflections across the coordinate planes,

$$L_{i,j} : (x, y, z) \mapsto \left( (-1)^i x, (-1)^j y, z \right), \forall i, j \in \{0, 1\}. \tag{47}$$

This HHD not only uncovers more knowledge of the Chua circuit's symmetry but can also be used to analysis the energy of the system. According to the generalized Hamiltonian formalism (Eq. 20), the energy function $H$ associated with the Chua circuit satisfies the following PDE:

$$\alpha y \frac{\partial H}{\partial x} + (x + z) \frac{\partial H}{\partial y} - \beta y \frac{\partial H}{\partial z} = 0, \tag{48}$$

which is satisfied by the quadratic form:

$$H = \frac{1}{2} \left( -\frac{1}{\alpha} x^2 + y^2 + \frac{1}{\beta} z^2 \right). \tag{49}$$

As presented by [77], this energy function indicates that the Chua circuit keeps oscillatory when the energy release along the $x$-axis is enough to balance the energy pumping along the $y$-axis and $z$-axis.

## F  Evaluation metrics for learning ODEs

**RMSE and MNLL of state derivatives**   To evaluate the models' performance in terms of learning $\dot{\mathbf{x}} = \mathbf{f}(\mathbf{x})$, we first compute the *root mean squared error* (RMSE) between the predicted time derivatives $\hat{\dot{\mathbf{x}}}_i$ and the ground truth $\dot{\mathbf{x}}_i$, *i.e.*

$$\text{RMSE} = \left( \frac{1}{m} \sum_{i=1}^{m} \| \hat{\dot{\mathbf{x}}}_i - \dot{\mathbf{x}}_i \|^2 \right)^{\frac{1}{2}}, \tag{50}$$

where $\|\cdot\|^2$ is the Euclidean norm, and $m$ is the number of test data. The lower the RMSE, the better. The test set $\{(\mathbf{x}_i, \dot{\mathbf{x}}_i)\}_{i=1}^{m}$ was generated by sampling a grid of points with the resolution of 10. In addition to the RMSE for evaluating the regression results, we further use the *mean negative log likelihood* (MNLL) to evaluate the prediction uncertainty provided by the GP models, and it is calculated by

$$\text{MNLL} = -\frac{1}{m} \sum_{i=1}^{m} \log \mathcal{N}(\dot{\mathbf{x}}_i \mid \hat{\dot{\mathbf{x}}}_i, var(\mathbf{x}_i)), \tag{51}$$

where $var(\mathbf{x}_i)$ is the prediction variance at a test state $\mathbf{x}_i$. The lower the MNLL, the more effectively the forecast uncertainty reflects the prediction error.

**VPT of state trajectories**   To obtain a more comprehensive evaluation of the learned ODE models, we considered additional metrics for evaluating the accuracy of predicting state trajectories. Recent studies [48, 30, 70] have suggested that RMSEs of state trajectories over long time horizons can be misleading indicators. So we used the *valid prediction time* (VPT) to measure the model's ability to do such long-term extrapolation in a phase space. Following the definition in [30, 70], VPT calculates the first time step $t$ at which the *normalized root mean square error* (NRMSE) between the predicted state $\hat{\mathbf{x}}_t$ and the ground truth $\mathbf{x}_t$ exceeds a given threshold $\epsilon$,

$$\text{VPT} = \frac{1}{T} \arg\min_t \{ \text{NRMSE}(\hat{\mathbf{x}}_t, \mathbf{x}_t) > \epsilon, \forall\, 0 \le t \le T \}, \tag{52}$$

where $\text{NRMSE}(\hat{\mathbf{x}}_t, \mathbf{x}_t) = \left( (\hat{\mathbf{x}}_t - \mathbf{x}_t)^\mathsf{T} \Sigma (\hat{\mathbf{x}}_t - \mathbf{x}_t) / n \right)^{\frac{1}{2}}$, $\Sigma = \text{diag}(1/\sigma_1, \ldots, 1/\sigma_n)$, with $\sigma_i$ denoting the variance of the $i$-th dimension of the true trajectory. The VPT measures how long the predicted trajectory remains close to the true trajectory, so the higher this indicator, the better. In the experiments, we set $\epsilon = 0.01$. To alleviate the dependency on the initial condition, we reported the VPT averaged over trajectories simulated from 50 randomly sampled initial conditions. And a trajectory from an initial condition was solved by the Dormand–Prince method (dopri5) [15] implemented in *torchdiffeq*[4], integrating forward in time at a frequency of 25 Hz for 15 seconds, with the relative and absolute tolerances of $10^{-6}$.

---

[4] https://github.com/rtqichen/torchdiffeq

# G Implementation details

The experiments were performed on a single Nvidia GeForce GTX 3050 Ti GPU, and all of the models were implemented with PyTorch [51]. The GP-based models (Ind-GP, GIM-GP, Div-GP, HHD-GP and SPHHD-GP) were trained by maximizing the log of their marginal likelihood:

$$\log p\left(\mathbf{Y} \mid \mathbf{X}\right) = \log \mathcal{N}\left(\mathbf{Y} \mid \mathbf{0}, \mathbf{K} + \Sigma\right) = -\frac{1}{2}\mathbf{Y}^{\mathsf{T}}\left(\mathbf{K} + \Sigma\right)^{-1}\mathbf{Y} - \frac{1}{2}\log|\mathbf{K} + \Sigma| - m\log 2\pi, \tag{53}$$

where $|\cdot|$ computes the determinant of the covariance matrix $\mathbf{K} + \Sigma$. Training a GP model refers to optimizing the kernel parameters in $\mathbf{K}$ and the noise variances in $\Sigma$, and these hyperparameters were initialized randomly in our experiments. The GP-based models were trained by the ADAM optimizer [34], with a learning rate 0.01 for 3000 gradient steps.

## G.1 Ind-GP

The Ind-GP models each dimension of a $n$-dimensional dynamical system independently, so its matrix-valued kernel is constructed by

$$\kappa_{ind} = \mathrm{diag}\left(\kappa_1, \ldots, \kappa_n\right), \tag{54}$$

where $\kappa_i$, $i = 1, \ldots, n$ are independent scalar kernels. And a standard choice for $\kappa_i$ is the squared exponential (SE) kernel,

$$\kappa_{se}\left(\mathbf{x}, \mathbf{x}'\right) = \sigma^2 \exp\left(-\frac{1}{2}l^{-2}\left\|\mathbf{x} - \mathbf{x}'\right\|^2\right), \tag{55}$$

which has two parameters: $\sigma$ determines the variation of function values from their mean, and $l$ controls the length scale on which the function varies. Realizations of GPs with SE kernels are dense in the set of smooth functions $C^\infty\left(\mathbb{R}^n, \mathbb{R}\right)$ (cf. Prop.1 in [37]). For a fair comparison, we also used the SE kernel (Eq. 55) to construct kernels for the other GP-based models.

## G.2 GIM-GP

The GIM-GP produces predictions with the desired symmetry. Each of the systems in our experiments exhibits the odd symmetry as a whole. So their symmetry groups can be given by $\mathcal{G}_{odd} = \{I_n, -I_n\}$, where $I_n$ is the $n$-dimensional identity matrix, and the group elements are linear representations of the group actions, $i.e.$, $L_g\left(\mathbf{x}\right) = g\mathbf{x}$, $\forall g \in \mathcal{G}$. Therefore, according to Eq. 17, the GIM-kernel for $\mathcal{G}_{odd}$ was constructed by

$$\kappa_{gim} = \left(\kappa_{se}\left(\mathbf{x}, \mathbf{x}'\right) - \kappa_{se}\left(\mathbf{x}, -\mathbf{x}'\right)\right) I_n, \tag{56}$$

where the SE kernel (Eq. 55) was used as the basis kernel $\kappa$ in Eq. 17.

## G.3 Div-GP

GPs with a div-free kernel can be used to approximate conservative dynamics. According to Eq. 12, a two-dimensional div-free kernel was constructed by

$$\kappa_{div} = \begin{bmatrix} \frac{\partial^2 \kappa_H\left(\mathbf{x}, \mathbf{x}'\right)}{\partial x_2 \partial x_2'} & -\frac{\partial^2 \kappa_H\left(\mathbf{x}, \mathbf{x}'\right)}{\partial x_2 \partial x_1'} \\ -\frac{\partial^2 \kappa_H\left(\mathbf{x}, \mathbf{x}'\right)}{\partial x_1 \partial x_2'} & \frac{\partial^2 \kappa_H\left(\mathbf{x}, \mathbf{x}'\right)}{\partial x_1 \partial x_1'} \end{bmatrix}, \tag{57}$$

where $\kappa_{\mathbf{u}}$ in Eq. 12 is denoted by $\kappa_H$ instead, indicating that it is the kernel for the Hamiltonian functions of the damped mass spring and the damped pendulum, and the SE kernel (Eq. 55) was used for $\kappa_H$. The partial derivatives in Eq. 57 were calculated by automatic differentiation in PyTorch, so we did not need to derive its analytic expression.

Then we constructed the div-free kernel for the Chua circuit system. According to Eq. 11, a three-dimensional div-free vector field is given by $\mathbf{f}_{div}\left(\mathbf{x}\right) = \Psi \mathbf{u}\left(\mathbf{x}\right)$, where $\Psi = [\psi_{12}, \psi_{13}, \psi_{23}] \in R^{3 \times 3}$

with its components given by

$$\psi_{12} = \begin{bmatrix} 0 & 1 & 0 \\ -1 & 0 & 0 \\ 0 & 0 & 0 \end{bmatrix} \begin{bmatrix} \partial_{x_1} \\ \partial_{x_2} \\ \partial_{x_3} \end{bmatrix} = \begin{bmatrix} \partial_{x_2} \\ -\partial_{x_1} \\ 0 \end{bmatrix}, \tag{58}$$

$$\psi_{13} = \begin{bmatrix} 0 & 0 & 1 \\ 0 & 0 & 0 \\ -1 & 0 & 0 \end{bmatrix} \begin{bmatrix} \partial_{x_1} \\ \partial_{x_2} \\ \partial_{x_3} \end{bmatrix} = \begin{bmatrix} \partial_{x_3} \\ 0 \\ -\partial_{x_1} \end{bmatrix}, \tag{59}$$

$$\psi_{23} = \begin{bmatrix} 0 & 0 & 0 \\ 0 & 0 & 1 \\ 0 & -1 & 0 \end{bmatrix} \begin{bmatrix} \partial_{x_1} \\ \partial_{x_2} \\ \partial_{x_3} \end{bmatrix} = \begin{bmatrix} 0 \\ \partial_{x_3} \\ -\partial_{x_2} \end{bmatrix}. \tag{60}$$

Then by assuming $\mathbf{u} = [u_{12}, u_{13}, u_{23}]^{\mathsf{T}} \sim \mathcal{GP}\left(\mathbf{0}, \kappa_{\mathbf{u}} = \kappa_{ind}\right)$, we constructed the div-free kernel for the Chua circuit according to Eq. 12:

$$\kappa_{div} = \begin{bmatrix} \partial_{x_2} & \partial_{x_3} & 0 \\ -\partial_{x_1} & 0 & \partial_{x_3} \\ 0 & -\partial_{x_1} & -\partial_{x_2} \end{bmatrix} \begin{bmatrix} \kappa_1 & 0 & 0 \\ 0 & \kappa_2 & 0 \\ 0 & 0 & \kappa_3 \end{bmatrix} \begin{bmatrix} \partial_{x_2} & -\partial_{x_1} & 0 \\ \partial_{x_3} & 0 & -\partial_{x_1} \\ 0 & \partial_{x_3} & -\partial_{x_2} \end{bmatrix} \tag{61}$$

$$= \begin{bmatrix} \frac{\partial^2 \kappa_1\left(\mathbf{x},\mathbf{x}'\right)}{\partial x_2 \partial x_2'} + \frac{\partial^2 \kappa_2\left(\mathbf{x},\mathbf{x}'\right)}{\partial x_3 \partial x_3'} & -\frac{\partial^2 \kappa_1\left(\mathbf{x},\mathbf{x}'\right)}{\partial x_2 \partial x_1'} & -\frac{\partial^2 \kappa_2\left(\mathbf{x},\mathbf{x}'\right)}{\partial x_3 \partial x_1'} \\ -\frac{\partial^2 \kappa_1\left(\mathbf{x},\mathbf{x}'\right)}{\partial x_1 \partial x_2'} & \frac{\partial^2 \kappa_1\left(\mathbf{x},\mathbf{x}'\right)}{\partial x_1 \partial x_1'} + \frac{\partial^2 \kappa_3\left(\mathbf{x},\mathbf{x}'\right)}{\partial x_3 \partial x_3'} & -\frac{\partial^2 \kappa_3\left(\mathbf{x},\mathbf{x}'\right)}{\partial x_3 \partial x_2'} \\ -\frac{\partial^2 \kappa_2\left(\mathbf{x},\mathbf{x}'\right)}{\partial x_1 \partial x_3'} & -\frac{\partial^2 \kappa_3\left(\mathbf{x},\mathbf{x}'\right)}{\partial x_2 \partial x_3'} & \frac{\partial^2 \kappa_2\left(\mathbf{x},\mathbf{x}'\right)}{\partial x_1 \partial x_1'} + \frac{\partial^2 \kappa_3\left(\mathbf{x},\mathbf{x}'\right)}{\partial x_2 \partial x_2'} \end{bmatrix}, \tag{62}$$

where $\kappa_1$, $\kappa_2$ and $\kappa_3$ are all independent SE kernels (Eq. 55).

## G.4 HHD-GP

The HHD-GP consists of two independent GPs added together, modeling curl-free and div-free dynamics respectively. GPs with div-free kernels have been constructed earlier, so here we build curl-free kernels according to Eq. 8. For the damped mass spring and the damped pendulum, their two-dimensional curl-free kernel was constructed by

$$\kappa_{curl} = \begin{bmatrix} \frac{\partial^2 \kappa_V\left(\mathbf{x},\mathbf{x}'\right)}{\partial x_1 \partial x_1'} & \frac{\partial^2 \kappa_V\left(\mathbf{x},\mathbf{x}'\right)}{\partial x_1 \partial x_2'} \\ \frac{\partial^2 \kappa_V\left(\mathbf{x},\mathbf{x}'\right)}{\partial x_2 \partial x_1'} & \frac{\partial^2 \kappa_V\left(\mathbf{x},\mathbf{x}'\right)}{\partial x_2 \partial x_2'} \end{bmatrix}. \tag{63}$$

Similarly, the three-dimensional curl-free kernel for the Chua circuit was constructed by

$$\kappa_{curl} = \begin{bmatrix} \frac{\partial^2 \kappa_V\left(\mathbf{x},\mathbf{x}'\right)}{\partial x_1 \partial x_1'} & \frac{\partial^2 \kappa_V\left(\mathbf{x},\mathbf{x}'\right)}{\partial x_1 \partial x_2'} & \frac{\partial^2 \kappa_V\left(\mathbf{x},\mathbf{x}'\right)}{\partial x_1 \partial x_3'} \\ \frac{\partial^2 \kappa_V\left(\mathbf{x},\mathbf{x}'\right)}{\partial x_2 \partial x_1'} & \frac{\partial^2 \kappa_V\left(\mathbf{x},\mathbf{x}'\right)}{\partial x_2 \partial x_2'} & \frac{\partial^2 \kappa_V\left(\mathbf{x},\mathbf{x}'\right)}{\partial x_2 \partial x_3'} \\ \frac{\partial^2 \kappa_V\left(\mathbf{x},\mathbf{x}'\right)}{\partial x_3 \partial x_1'} & \frac{\partial^2 \kappa_V\left(\mathbf{x},\mathbf{x}'\right)}{\partial x_3 \partial x_2'} & \frac{\partial^2 \kappa_V\left(\mathbf{x},\mathbf{x}'\right)}{\partial x_3 \partial x_3'} \end{bmatrix}, \tag{64}$$

where $\kappa_V$ is a SE kernel.

## G.5 SPHHD-GP

### G.5.1 Symmetry-preserving curl-free kernels

According to Theorem 5.2, a $\mathcal{G}$-equivariant curl-free vector field is constructed by constraining its scalar potential $V$ to be $\mathcal{G}$-invariant. Therefore, a $\mathcal{G}$-equivariant curl-free kernel is obtained by constructing its potential kernel $\kappa_V$ according to Eq. 16.

The curl-free vector field of the damped mass spring and the damped pendulum has two types of symmetry:

*(i)* translation along the $q$-axis, *i.e.*, $L_g\left(\mathbf{x}\right) = \mathbf{x} + (g, 0)$, for all $g \in \mathbb{R}$;

*(ii)* rotation by an angle of $\pi$ radians, *i.e.*, $L_g\left(\mathbf{x}\right) = g\mathbf{x}$, for all $g \in \{I_2, -I_2\}$.

These symmetry groups were enforced to a SE kernel sequentially according to Eq. 16, where $|\mathcal{G}|$ can be ignored. Specifically, the translation invariance was enforced using the Gaussian integral formula:

$$\kappa_V = \int_{-\infty}^{+\infty} \kappa_{se}\left(\mathbf{x}, \mathbf{x}' + [g, 0]^\mathsf{T}\right) dg \tag{65}$$

$$= \kappa_{se}\left(p, p'\right) \int_{-\infty}^{+\infty} \exp(-\frac{(q - q' - g)^2}{2l^2}) dg \tag{66}$$

$$= \sqrt{2\pi} l \kappa_{se}\left(p, p'\right), \tag{67}$$

then based on which the $\pi$ rotation invariance was enforced by

$$\kappa_V = \sqrt{2\pi} l \left(\kappa_{se}\left(p, p'\right) + \kappa_{se}\left(p, -p'\right)\right). \tag{68}$$

The curl-free vector field of the Chua circuit (Eq. 46) has similar two types of symmetry:

*(i)* translation along the $z$-axis, *i.e.*, $L_g\left(\mathbf{x}\right) = \mathbf{x} + (0, 0, g)$, for all $g \in \mathbb{R}$;

*(ii)* mirror reflections across the coordinate planes, *i.e.*, $L_{i,j} : (x, y, z) \mapsto \left((-1)^i x, (-1)^j y, z\right)$, for all $i, j \in \{0, 1\}$.

Therefore, $\kappa_V$ for the Chua circuit was constructed by

$$\kappa_V = \sqrt{2\pi} l \sum_{g \in \mathcal{G}} \kappa_{se}\left(\mathbf{x}, g\mathbf{x}'\right), \quad \mathcal{G} = \left\{\mathrm{diag}\left((-1)^i, (-1)^j, 0\right) \mid i, j \in \{0, 1\}\right\}. \tag{69}$$

### G.5.2 Symmetry-preserving div-free kernels

Theorem 5.3 shows that a $\mathcal{G}$-equivariant div-free vector field is constructed from a vector potential $\mathbf{v}$ with the same equivariance. For the div-free vector field of the damped mass-spring system with SO(2)-equivariance, its $\kappa_\mathbf{v}$ was constructed by the GIM-kernel (Eq. 17):

$$\kappa_\mathbf{v}\left(\mathbf{x}, \mathbf{x}'\right) = \int_0^{2\pi} \kappa_{se}\left(\mathbf{x}, g_\theta \mathbf{x}'\right) g_\theta d\theta, \text{ where } g_\theta = \begin{bmatrix} \cos\theta & -\sin\theta \\ \sin\theta & \cos\theta \end{bmatrix}. \tag{70}$$

Then this $\kappa_\mathbf{v}$ (Eq. 70) is substituted into Eq. 18 to construct $\kappa_H$ for its div-free kernel $\kappa_{div}$ (Eq. 57):

$$\kappa_H = \frac{\partial^2}{\partial x_2 \partial x_2'} [\kappa_\mathbf{v}]_{1,1} + \frac{\partial^2}{\partial x_1 \partial x_1'} [\kappa_\mathbf{v}]_{2,2} - \frac{\partial^2}{\partial x_2 \partial x_1'} [\kappa_\mathbf{v}]_{1,2} - \frac{\partial^2}{\partial x_1 \partial x_2'} [\kappa_\mathbf{v}]_{2,1} \tag{71}$$

$$= \int_0^{2\pi} \kappa_{se}\left(\mathbf{x}, g_\theta \mathbf{x}'\right) \frac{2l^2 - \|\mathbf{x} - g_\theta \mathbf{x}'\|^2}{l^4} d\theta. \tag{72}$$

which admits no closed-form solution, so we used a numerical approximation of the integral (Eq. 72) by sampling discrete rotations of $\left\{\theta = \frac{\pi}{4} n \mid n = 0, \dots, 7\right\}$, whose rotation matrices form a finite group. And by setting $\theta \in \{0, \pi\}$, Eq. 72 was used to construct $\kappa_H$ for the div-free vector field of the damped pendulum, which has the equivariance under a $\pi$ rotation (odd symmetry).

The div-free vector field of the Chui circuit (Eq. 46) has the odd symmetry and the translation symmetry along $x = z$. $\kappa_\mathbf{v}$ respecting the translation symmetry was constructed by $\kappa_\mathbf{v} = \kappa \cdot I_3$, where $\kappa$ is given by

$$\kappa = \int_{-\infty}^{+\infty} \kappa_{se}\left([x, y, z]^\mathsf{T}, [x', y', z']^\mathsf{T} + [g, 0, -g]^\mathsf{T}\right) dg \tag{73}$$

$$= \kappa_{se}\left(y, y'\right) \int_{-\infty}^{+\infty} \exp(-\frac{(x - x' - g)^2 + (z - z' + g)^2}{2l^2}) dg \tag{74}$$

$$= \kappa_{se}\left(y, y'\right) \int_{-\infty}^{+\infty} \exp\left(-\frac{2\left(g - \frac{x_1 - x_1' - x_3 + x_3'}{2}\right)^2 + \frac{(x_1 - x_1' + x_3 - x_3')^2}{2}}{2l^2}\right) dg \tag{75}$$

$$= \sqrt{\pi} l \kappa_{se}\left(x_2, x_2'\right) \int_{-\infty}^{+\infty} \exp\left(-\frac{(x_1 - x_1' + x_3 - x_3')^2}{4l^2}\right) dg \tag{76}$$

$$= \sqrt{\pi} l \kappa_{se}\left(\mathbf{p}, \mathbf{p}'\right). \tag{77}$$

where $\mathbf{p} = \left[ \frac{1}{\sqrt{2}} \left( x + z \right), y \right]^{\mathsf{T}}$. And we further enforced the odd symmetry by

$$\kappa = \sqrt{\pi} l \left( \kappa_{se} \left( \mathbf{p}, \mathbf{p}' \right) - \kappa_{se} \left( \mathbf{p}, -\mathbf{p}' \right) \right). \tag{78}$$

Then, $\kappa_{\mathbf{v}} = \kappa \cdot I_3$ was substituted into Eq. 18 to construct $\kappa_{\mathbf{u}}$ by

$$\kappa_{\mathbf{u}} = \begin{bmatrix} \frac{\partial^2 \kappa(\mathbf{x},\mathbf{x}')}{\partial x_2 \partial x_2'} + \frac{\partial^2 \kappa(\mathbf{x},\mathbf{x}')}{\partial x_1 \partial x_1'} & \frac{\partial^2 \kappa(\mathbf{x},\mathbf{x}')}{\partial x_2 \partial x_3'} & -\frac{\partial^2 \kappa(\mathbf{x},\mathbf{x}')}{\partial x_1 \partial x_3'} \\ \frac{\partial^2 \kappa(\mathbf{x},\mathbf{x}')}{\partial x_3 \partial x_2'} & \frac{\partial^2 \kappa(\mathbf{x},\mathbf{x}')}{\partial x_3 \partial x_3'} + \frac{\partial^2 \kappa(\mathbf{x},\mathbf{x}')}{\partial x_1 \partial x_1'} & \frac{\partial^2 \kappa(\mathbf{x},\mathbf{x}')}{\partial x_1 \partial x_2'} \\ -\frac{\partial^2 \kappa(\mathbf{x},\mathbf{x}')}{\partial x_3 \partial x_1'} & \frac{\partial^2 \kappa(\mathbf{x},\mathbf{x}')}{\partial x_2 \partial x_1'} & \frac{\partial^2 \kappa(\mathbf{x},\mathbf{x}')}{\partial x_3 \partial x_3'} + \frac{\partial^2 \kappa(\mathbf{x},\mathbf{x}')}{\partial x_2 \partial x_2'} \end{bmatrix}. \tag{79}$$

Finally, this $\kappa_{\mathbf{u}}$ (Eq. 79) was substituted into Eq. 12 to construct the symmetry-preserving div-free kernel for the Chua circuit.

## G.6 D-HNN

We used the released code[5] of D-HNN and ran their training routine for our systems.

## G.7 Implementation details of predicting energy

### G.7.1 Predicting energy of the mass-spring system and the pendulum

When HHD-GP and SPHHD-GP are used to learn the damped mass-spring system and the damped pendulum, the potential kernel $\kappa_H$ for constructing their div-free kernels (Eq. 57) can be interpreted as placing a GP prior on the Hamiltonian function. Therefore, a joint GP describing both the Hamiltonian function $H\left(\mathbf{x}\right)$ and the dynamical system $\mathbf{f}\left(\mathbf{x}\right)$ is given by

$$\begin{bmatrix} H\left(\mathbf{x}\right) \\ \mathbf{f}\left(\mathbf{x}\right) \end{bmatrix} \sim \mathcal{GP} \left( \begin{bmatrix} 0 \\ \mathbf{0} \end{bmatrix}, \begin{bmatrix} \kappa_H\left(\mathbf{x},\mathbf{x}'\right) & \kappa_{H,\mathbf{f}}\left(\mathbf{x},\mathbf{x}'\right) \\ \kappa_{\mathbf{f},H}\left(\mathbf{x},\mathbf{x}'\right) & \kappa_{hhd}\left(\mathbf{x},\mathbf{x}'\right) \end{bmatrix} \right), \tag{80}$$

where $\kappa_{H,\mathbf{f}}\left(\mathbf{x},\mathbf{x}'\right) = \kappa_{\mathbf{f},H}\left(\mathbf{x}',\mathbf{x}\right)^{\mathsf{T}}$ with

$$\kappa_{H,\mathbf{f}}\left(\mathbf{x},\mathbf{x}'\right) = \mathrm{cov}\left[H\left(\mathbf{x}\right), \mathbf{f}\left(\mathbf{x}'\right)\right] = \left[\partial_{p'}, -\partial_{q'}\right] \kappa_H\left(\mathbf{x},\mathbf{x}'\right); \tag{81}$$

$$\kappa_{\mathbf{f},H}\left(\mathbf{x},\mathbf{x}'\right) = \mathrm{cov}\left[\mathbf{f}\left(\mathbf{x}\right), H\left(\mathbf{x}'\right)\right] = \left[\partial_p, -\partial_q\right]^{\mathsf{T}} \kappa_H\left(\mathbf{x},\mathbf{x}'\right). \tag{82}$$

After training the model $\mathbf{f}\left(\mathbf{x}\right) \sim \mathcal{GP}\left(\mathbf{0}, \kappa_{hhd}\left(\mathbf{x},\mathbf{x}'\right)\right)$ on noisy observations $\mathbf{Y} = \left[\mathbf{y}_1, \ldots, \mathbf{y}_m\right]^{\mathsf{T}}$[6] at states $\mathbf{X} = \left[\mathbf{x}_1, \ldots, \mathbf{x}_m\right]^{\mathsf{T}}$, we are interested in predicting the value of Hamiltonian function $H\left(\mathbf{x}_*\right)$ at a new test state $\mathbf{x}_*$. Since these data determine $H\left(\cdot\right)$ only up to an additive constant, we assume that we have an anchor point $H\left(\mathbf{x}_0\right)$ which can be chosen arbitrarily. Then, according to the GP prior (Eq. 80), $H\left(\mathbf{x}_*\right)$ and $\mathbf{Y}_H = \left[H\left(\mathbf{x}_0\right), \mathbf{Y}\right]^{\mathsf{T}}$ are jointly distributed as

$$\begin{bmatrix} H\left(\mathbf{x}_*\right) \\ \mathbf{Y}_H \end{bmatrix} \sim \mathcal{N} \left( \begin{bmatrix} 0 \\ \mathbf{0} \end{bmatrix}, \begin{bmatrix} \kappa_H\left(\mathbf{x}_*, \mathbf{x}_*\right) & \mathbf{k} \\ \mathbf{k}^{\mathsf{T}} & \mathbf{K} \end{bmatrix} \right), \tag{83}$$

where $\mathbf{k} = \left[\kappa_H\left(\mathbf{x}_*, \mathbf{x}_0\right), \kappa_{H,\mathbf{f}}\left(\mathbf{x}_*, \mathbf{X}\right)\right]$, and

$$\mathbf{K} = \begin{bmatrix} \kappa_H\left(\mathbf{x}_0, \mathbf{x}_0\right) & \kappa_{H,\mathbf{f}}\left(\mathbf{x}_0, \mathbf{X}\right) \\ \kappa_{\mathbf{f},H}\left(\mathbf{X}, \mathbf{x}_0\right) & \kappa_{hhd}\left(\mathbf{X}, \mathbf{X}\right) + \sigma I \end{bmatrix}. \tag{84}$$

Then, we obtain the posterior distribution

$$p\left(H\left(\mathbf{x}_*\right) \mid \mathbf{Y}_H\right) = \mathcal{N}\left(\mathbf{k}\mathbf{K}^{-1}\mathbf{Y}_H, \; \kappa_H\left(\mathbf{x}_*, \mathbf{x}_*\right) - \mathbf{k}\mathbf{K}^{-1}\mathbf{k}^{\mathsf{T}}\right), \tag{85}$$

where the mean function is used for energy prediction.

---

[5] https://github.com/DrewSosa/dissipative_hnns

[6] $\mathbf{y}_i = \mathbf{f}\left(\mathbf{x}_i\right) + \epsilon, \epsilon \overset{\text{i.i.d}}{\sim} \mathcal{N}\left(\mathbf{0}, \sigma I\right)$

### G.7.2 Predicting energy of the Chua circuit

[60] stated that the div-free component of a chaotic system is always orthogonal to the gradient of its energy function, $i.e.$, $\nabla H^\mathsf{T}\mathbf{f}_{div}(\mathbf{x}) = 0$, for all $\mathbf{x} \in \mathbb{R}^n$. The energy function of the Chua circuit is in a quadratic form:

$$H = \frac{1}{2}\left(-\frac{1}{\alpha}x^2 + y^2 + \frac{1}{\beta}z^2\right). \tag{86}$$

So we parameterize the energy function of the Chua circuit by $\hat{H}(\mathbf{x}) = \frac{1}{2}\left(a_1 x^2 + a_2 y^2 + a_3 z^2\right)$, where $\mathbf{a} = [a_1, a_2, a_3]$ are parameters. Then by learning a divergence-free vector field $\hat{\mathbf{f}}_{div}(\cdot)$ through our model, we can estimate the parameters $\mathbf{a}$ by minimizing $\sum_{i=1}^{m}\left(\nabla\hat{H}^\mathsf{T}\hat{\mathbf{f}}(\mathbf{x}_i)\right)^2$ at a finite number of sample points $\{\mathbf{x}_i\}_{i=1}^{m}$ ($m = 500$ in our experiments). And to eliminate the solution at $\mathbf{a} = [0, 0, 0]$, we add an equality constraint at a random point of the ground truth $\{\mathbf{x}_0, H(\mathbf{x}_0)\}$. Therefore, the parameters $\mathbf{a}$ of $\hat{H}$ are solved in a convex quadratic program (QP):

$$\begin{aligned}\min \quad & \mathbf{a}^T \mathbf{Q} \mathbf{a}\\ \text{s.t.} \quad & \hat{H}(\mathbf{x}_0) = H(\mathbf{x}_0)\end{aligned} \tag{87}$$

where $\mathbf{Q} = \mathbf{q}^\mathsf{T}\mathbf{q}$ with $\mathbf{q} = \left[\hat{\mathbf{f}}_{div}(\mathbf{x}_1), \ldots, \hat{\mathbf{f}}_{div}(\mathbf{x}_m)\right]^\mathsf{T} \in \mathbb{R}^{m\times 3}$. And in the experiments, we solved the QP using the solver provided by CVXOPT [2].

## H    Uniqueness and Symmetries

The two constituent kernels of our model define the space of divergence-free vector fields ($\mathbf{f}_{div} \in \mathcal{F}_{div}$) and the space of curl-free vector fields ($\mathbf{f}_{curl} \in \mathcal{F}_{curl}$), respectively. These two spaces overlap partially due to the presence of harmonic vector fields ($\mathbf{f}_{harm} \in \mathcal{F}_{div} \cap \mathcal{F}_{curl}$). To eliminate this overlap and make HHD unique, we propose to impose symmetry constraints on the two spaces separately, with the corresponding symmetry groups defined as $\mathcal{G}_{div}$ and $\mathcal{G}_{curl}$. Therefore, the uniqueness property of HHD depends on the space of harmonic vector fields that respects the union of two symmetry groups, $i.e.$

$$\mathcal{F}_{harm} = \left\{\mathbf{f}_{harm} \mid \mathbf{f}_{harm} \circ L_g = \mathbf{J}_{L_g}\mathbf{f}_{harm}, \forall\mathbf{x} \in \mathbb{R}^n, g \in \mathcal{G}_{div} \cup \mathcal{G}_{curl}\right\}. \tag{88}$$

If $\mathcal{F}_{harm} = \emptyset$, the HHD is unique, $i.e.$ our model is identifiable. A harmonic vector field can always be represented by the gradient field of a harmonic function $h$, which is a scalar function satisfying $\nabla \cdot \nabla h = 0$ (Laplace's equation). For the three dynamical systems used in our experiments, it can be easily verified that the presence of harmonic components can be eliminated through forced symmetries. The symmetry group union of a damped mass-spring system consists of a rotation group $\mathcal{G}_{div} = SO(2)$ and a translation group $\mathcal{G}_{curl} = \{(g, 0) \mid g \in \mathbb{R}\}$. A harmonic vector field $\mathbf{f}_{harm} = \nabla h$ that satisfies this translation symmetry implies that its harmonic function $h(q, p)$ is independent of the variable $q$, so the harmonic vector field is given by $\mathbf{f}_{harm} = (0, \partial_p h)$ and $\partial_p h$ should be a constant to satisfy the Laplace's equation, but the harmonic vector field in the form of constant clearly contradicts rotation symmetry $\mathcal{G}_{div}$. Therefore, there is no harmonic vector field that respects both the symmetry groups $\mathcal{G}_{div}$ and $\mathcal{G}_{curl}$. Similar conclusions can be drawn for the damped pendulum and the Chua circuit. The Laplace's equation and translation symmetry imply that harmonic vector fields can only exist in the form of constant vector fields. However, constant vector fields obviously contradict odd symmetry or mirror symmetry.

## I    Computational complexity

Compared to the diagonal kernel that independently models each dimension of a dynamical system [13, 31, 68, 69], our kernel can capture correlations among output dimensions because its off-diagonal elements are non-zero. However, this advantage comes together with a low computational efficiency. The exact inference of HHD-GP over an $n$-dimensional system with $m$ data points has a cost of $\mathcal{O}(m^3 n^3)$ in time, due to the inversion of its full covariance matrix $\mathbf{K}_* \in \mathbb{R}^{mn\times mn}$. Although in this work we do not focus on the computational challenge in large-scale applications, one can plug our kernel in a variety of sparse GP frameworks [62, 66], reducing the computational complexity to $\mathcal{O}(M^2 mn)$, where $M$ ($M \ll m$) is the number of inducing point to approximate the covariance matrix.

# J    Additional Experimental Results

## J.1    Results with increasing number of training data

To further compare the predictive performance of the models, we evaluated them by increasing the amount of training data. The results are shown in Fig. 5. As expected, the performance of the models improves as the amount of training data increases, but we can observe that our model (SPHHD-GP) performs best at every data amount relative to the baseline models, verifying the robustness of our model against data sparsity.

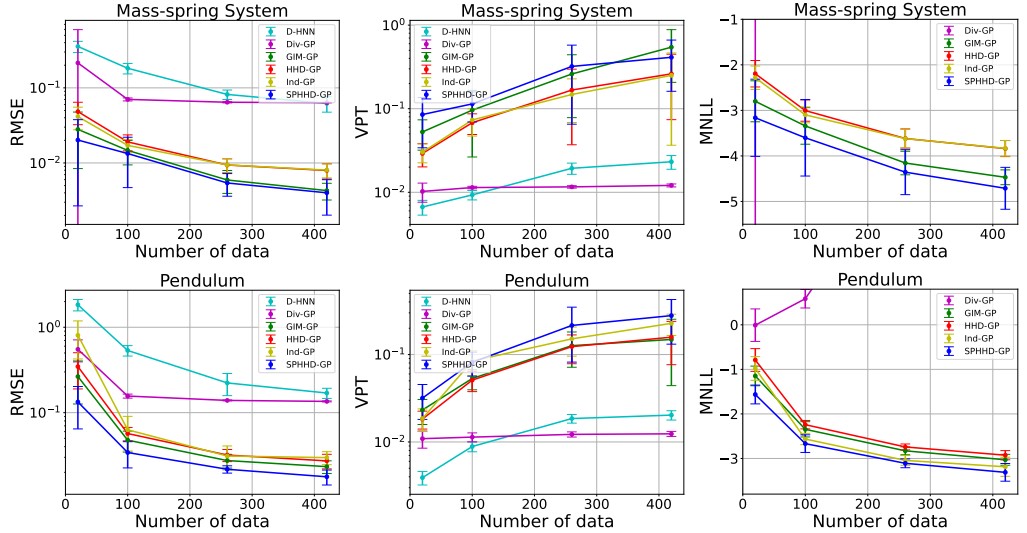

Figure 5: Results with an increasing number of training data (20, 100, 260, 420), which are averaged over 10 independent experiments performed by resampling the training sets and model initial parameters. The first and second columns are the results for the mass-spring system and the pendulum, respectively.

## J.2    Trajectory prediction of the mass-spring system and the pendulum

Fig. 6 and Fig. 7 present the trajectory predictions for the damped mass-spring system and the damped pendulum, respectively.

## J.3    Example Decomposition

Fig. 8 presents predicted decompositions of HHD-GP and SPHHD-GP. It can be observed that although they capture highly similar system dynamics (the first column in Fig. 8(a)), they learn completely different decompositions (the second and third columns in Fig. 8(a)). Compared with the ground truth in Fig. 1, SPHHD-GP learns the physically correct decomposition, so it can accurately predict the system energy. From Fig. 8(b) we can find that the predictions of HHD-GP have large variance, meaning that the model is less certain in isolating individual effects from other terms.

## J.4    Experiments of predicting dynamics with unseen frictions

To demonstrate the interpretability of the learned curl-free features, we adapted the trained model to predict dynamics with different friction coefficients. As detailed in Appendix A.1, the curl-free dynamics in the mass-spring system and the pendulum is caused by friction forces in the system. Utilizing this interpretation and the additive structure of the HHD-based models, we can generalize the models to perform inference over dynamics with different friction coefficients. We performed this experiment by first training HHD-GP, SPHHD-GP and D-HNN with data when $\rho = 0.1$ ($\rho$ is the friction coefficients), then generalizing the trained models to dynamics when $\rho = 0.05$ and $\rho = 0.5$ by multiplying the learned curl-free component with constants $0.5$ and $5.0$, respectively. The results

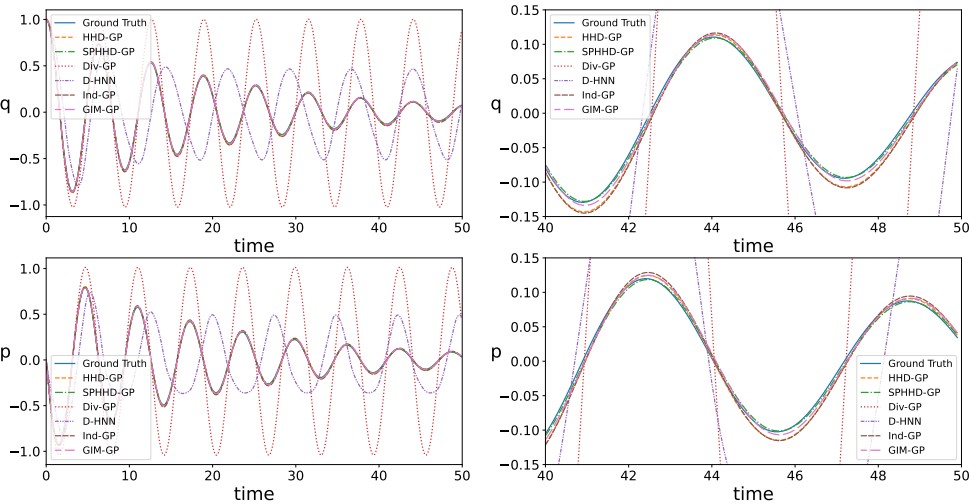

Figure 6: Comparison of predicted trajectories of the mass-spring system.

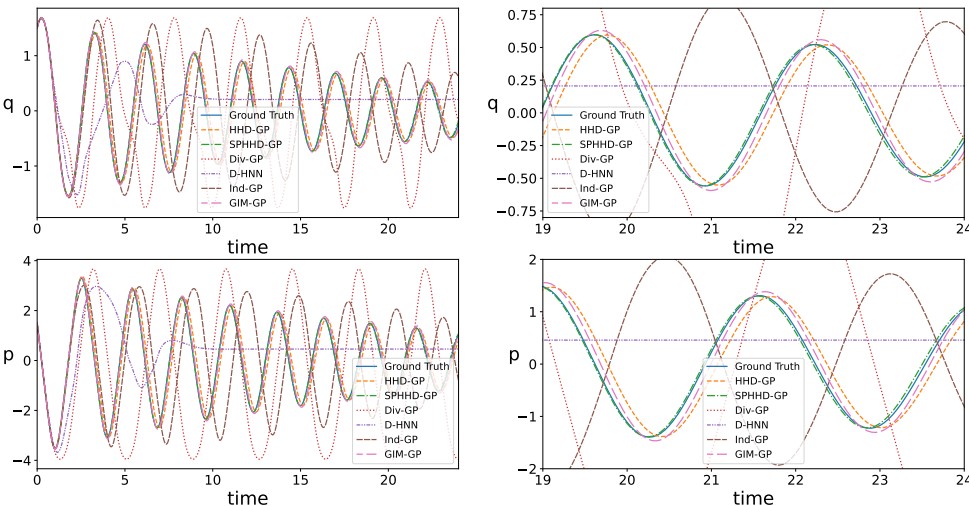

Figure 7: Comparison of predicted trajectories of the pendulum.

are shown in Fig. 9, where we can observe that the three models accurately predict the trajectories of the systems with a friction coefficient of 0.1 (first column in Fig. 9), but HHD-GP and D-HNN are difficult to generalize to cases with friction coefficients of 0.05 and 0.5 (second and third columns in Fig. 9). In contrast, SPHHD-GP effectively captures the dynamics of different friction conditions, even though it has not been trained for these specific friction coefficients. This adaptability is valuable because it allows the model to be applied to real-world scenarios where friction coefficients may change.

### J.5 Results of increasing noise level

To evaluate the robustness of the models, we added the experiments of increasing the standard deviation $\sigma = \{0.01, 0.05, 0.10, 0.20\}$ of Gaussian noise in the training data. Fig. 10 presents the results on the pendulum. As expected, the performance of the models degrades as noise increases. However, it can be observed that SPHHD-GP still performs best at all noise levels compared to the baselines, meaning that our model is more robust to data noise. Furthermore, from Fig. 10(d) we observe that the energy predictions of HHD-GP and D-HNN generate consistently significant errors, insensitive to noise level in training data, which again demonstrate the non-identifiability suffered by these models. In contrast, the energy errors of SPHHD-GP exhibit an increasing trend with increasing

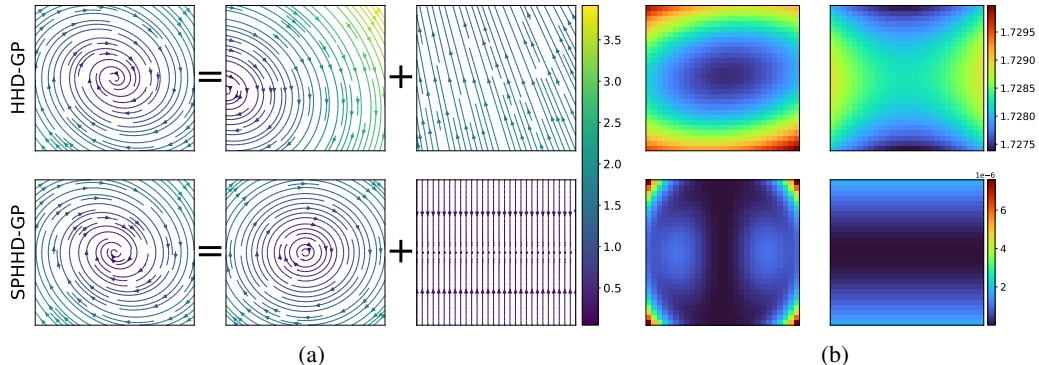

Figure 8: Example predicted HHD of the damped mass-spring system by HHD-GP (the first row) and SPHHD-GP (the second row). (a): predicted vector field (the first column) with its div-free (the second column) and curl-free (the third column) components; (b): the associated variance of the div-free (the first column) and curl-free (the second column) predictions.

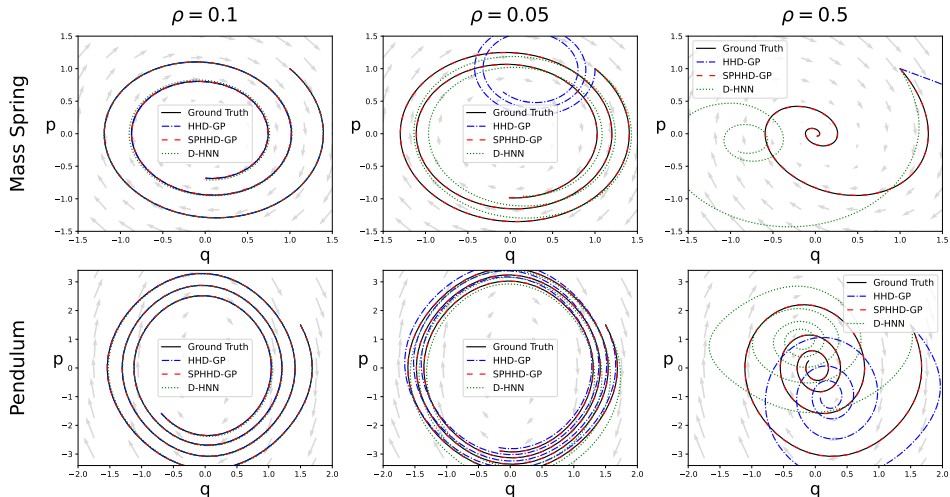

Figure 9: Adapting the trained models to predict the trajectories for different friction coefficients. The models are trained under $\rho = 0.1$, where $\rho$ is the friction coefficients, then they are generalized to predict trajectories for $\rho = 0.05$ and $\rho = 0.5$ by multiplying the learned curl-free component with constants $0.5$ and $5.0$, respectively. The lines in the figure represent predicted trajectories from different models (distinguished by different line styles), with the initial points for the two systems (the first and second rows) being $(1, 1)$ and $(1.5, 1.5)$, respectively. The background vector fields are the ground truth dynamics.

noise, further demonstrating the effectiveness of addressing model identifiability through symmetry constraints.

### J.6 Trajectory and energy prediction of the Chua circuit

Fig. 12 presents the comparison of trajectory predictions for the Chua circuit, and Fig. 11 compares the energy prediction along the trajectory.

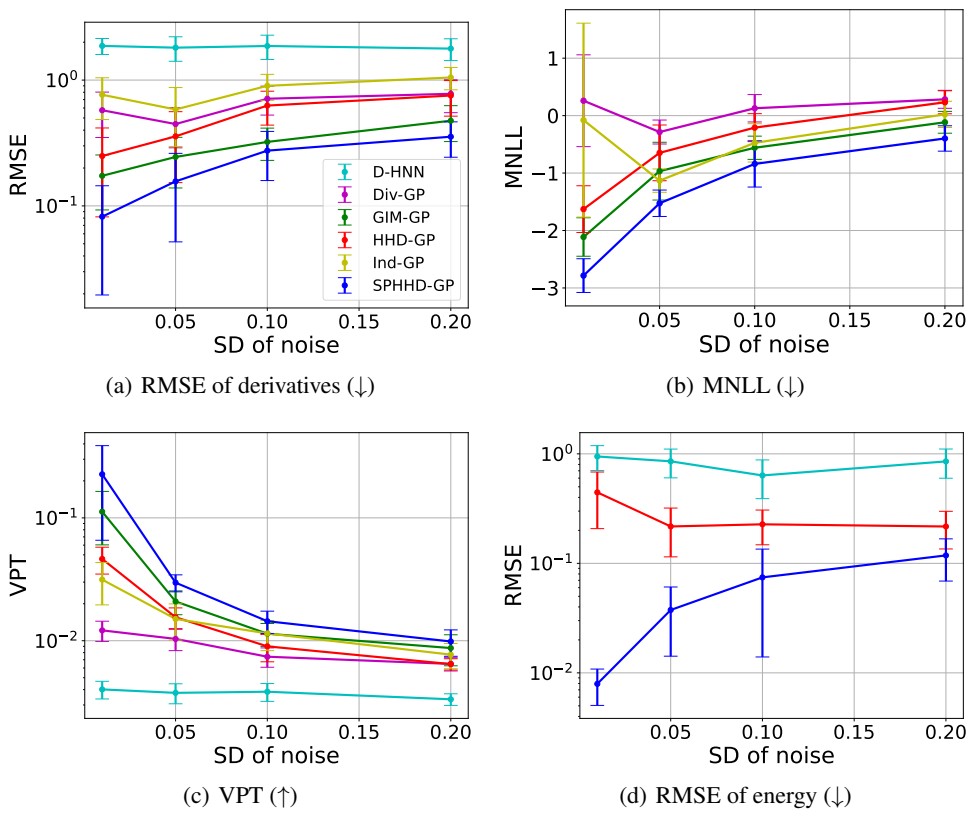

(a) RMSE of derivatives (↓)

(b) MNLL (↓)

(c) VPT (↑)

(d) RMSE of energy (↓)

Figure 10: Results with increasing standard deviation (SD) of Gaussian noise in training data.

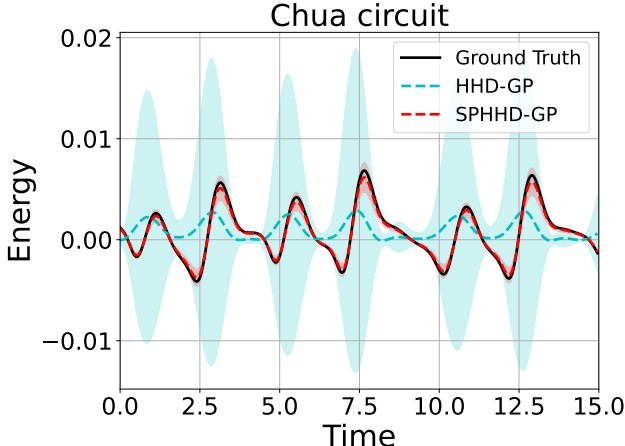

Figure 11: Comparison of predicted energy of the Chua circuit.

# K Societal impact

Generally, our model can have a significant impact on a wide range of applications, such as physical simulation and robotic control. However, it is imperative that we meticulously evaluate the performance of prediction and uncertainty estimation when implementing it in societal contexts.

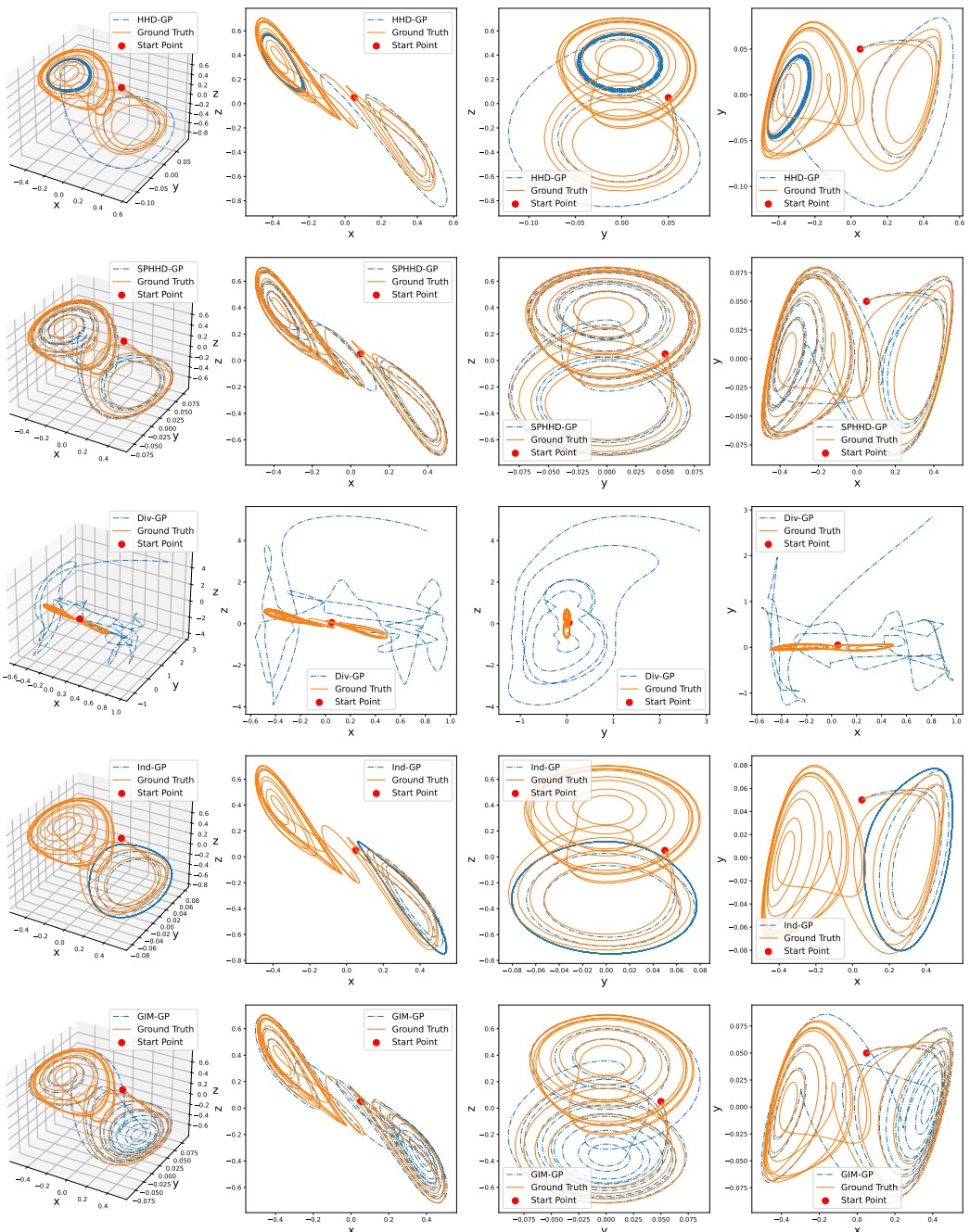

Figure 12: Comparison of predicted trajectories of the Chua circuit.

