# OpenReview forum: "HHD-GP: Incorporating Helmholtz-Hodge Decomposition into Gaussian Processes for Learning Dynamical Systems"
_NeurIPS.cc/2024/Conference — NeurIPS 2024 poster_

### Official Review · Reviewer_KheP · 2024-07-05

**Soundness:** 3
**Presentation:** 3
**Contribution:** 2
**Rating:** 6
**Confidence:** 3

**Summary:**

This paper proposes a novel dimensionality reduction method.
The method relies on the Helmholz-Hodge decomposition to identify the dynamical system through a decomposition into a curl-free and a divergence-free part.
Furthermore, the method introduces a way to incorporate priors that constrain the identified vector fields based on symmetry constraints.
This method is applied to various toy models (mass-spring system, pendulum, Chua circuit) and to real world data (ocean current fields).

**Strengths:**

This paper addresses an important problem in dynamical systems modeling.
This paper is the first to introduce the use of symmetries to construct unique GP models of dynamics that are decomposed into div- and curl-free vector fields.
This method is shown to be consistent in terms of convergence to the true parameter values as the amount of data increases.
The paper discusses the computational complexity of the proposed method.
The paper demonstrates that the presented framework is outperforms baselines on various toy models
Furthermore, the method is shown to be able to handle noisy data.
Finally, the method is applied to a real world dataset, which indicates practical significance.

**Weaknesses:**

The overall applicability of this method is questionable due to not completely addressing the identifiability issue of the decomposition. The issue is only resolved for systems with known symmetries, see also Questions.

The paper mentions that the accurate prediction of energy demonstrates the interpretability of learned div-free features.
However, this is not well-explained. The paper claims that the curl- and div-free components are physically meaningful, but this should be better justified.

The paper is missing comparison to other existing approaches for modeling physical systems. See for example:
Guo, Q., Mandal, M. K., \& Li, M. Y. (2005). Efficient Hodge–Helmholtz decomposition of motion fields. Pattern Recognition Letters, 26(4), 493-501.

Compare to other dynamical systems reconstruction methods:
- Hess, F., Monfared, Z., Brenner, M., \& Durstewitz, D. (2023). Generalized teacher forcing for learning chaotic dynamics. arXiv preprint arXiv:2306.04406.

-I. Mac\^edo and R. Castro, “Learning Divergence-Free and Curl-Free Vector Fields with Matrix-Valued Kernels,” technical report, Instituto de Matema ́tica Pura e Aplicada, Rio de Janeiro, Brazil, 2010.

Finally, there is no code provided to show the method's working and not all hyperparameters are described (see also Questions).

**Questions:**

What is the relevance of $\mathcal{G}$-equivariance and uniqueness/identifiability?
Appendix H: Uniqueness and Symmetries
- How do you know what priors to choose? Isn't an agnostic method better?
- How do you know which symmetries to enforce?

Could it be more fruitful to restrict yourself to a compact manifold with boundary so that the decomposition is unique?



# Implementation details
- How were the kernel parameters and noise variances initialized? Does training depend on initialization?
(Here you mention that these are hyperparameters, but typically one refers to non trained parameters as hyperparamters.)
-What were the ADAM optimizer paramters used for training?


# Choice of evaluation metrics
-Why did you mostly focus on performance of the vector field? If the point is learning the dynamics, metrics that look at the dynamics directly would be preferable.
Why not prediction of the future state/trajectory? Why prediction only measured through VPT?

**Limitations:**

The authors should comment on limitations and societal impact.

---

> ### Author Rebuttal · Authors · 2024-08-07
>
> # Response to Reviewer KheP
>
> We are very grateful for your valuable comments and acknowledgement of our main contributions. We sincerely appreciate your time and effort in reviewing our paper. Here is our response to your comments.
>
> ---
>
> > **Question 1**: How do you know what priors to choose? Isn't an agnostic method better? How do you know which symmetries to enforce? Could it be more fruitful to restrict yourself to a compact manifold with boundary so that the decomposition is unique?
>
> **Response**:
>
> The current work assumes that the prior of symmetries is directly available, and focuses on developing the method of incorperating symmetries into a GP model. This assumption is reasonable for many dynamical systems, but we totally agree with your concern about the availability of symmetry. Although physical systems always adhere to symmetries, the symmetries of many complex phenomena are often considered to be partially known or unknown. Now that our model has been demonstrated to perform well when system symmetry is available, we look forward to developing a method for symmetry-agnostic scenarios, i.e. enhancing our model with the capabilities of automatedly learning symmetries. Several recent studies have explored the feasibility of learning hidden symmetries from data [1, 2]. Since our model is developed in a probabilistic framework, another promising direction is to incorporate approximate symmetry instead. Studying the relationship between approximate symmetry and model identifiability is definitely an interesting and promising but unexplored research problem.
>
> We also agree with you that enforcing boundary condition is another effective way to make the decomposition unique. Our method of enforcing symmetry can also be used to impose the boundary condition, which is demonstrated by our experiments on learning ocean current fields (Section 6.3 of the manuscript). In this experiment, mirror symmetry was incorporated into the div-free kernel to enforce the divergence-free vector field to be parallel to the domain boundary, which is a sufficient condition for the decomposition to be unique [3]. The experimental results in Section 6.3 of the manuscript demonstrate that enforcing the boundary condition allows our model to provide more realistic ocean current predictions and divergence identification than the baselines.
>
> > **Question 2**: How were the kernel parameters and noise variances initialized? Does training depend on initialization? What were the ADAM optimizer paramters used for training?
>
> **Response**: For the experiments on simulated systems (damped mass-spring system, damped pendulum, and Chua circuit), we set the noise level to 0.05 (lines 275 and 355 in the main text). To explore the impact of noise level on model performance, in Appendix J.5 we provide experimental results with increasing noise in the training data (0.01, 0.05, 0.1, 0.2). For the experiments on a realistic system (ocean current field), we did not manually add noise to the training data since we used a realistic dataset. The kernel parameters are initialized randomly from a range of [0.1, 10]. In Table 1 of the manuscript we reported experimental results averaged over 10 independent experiments performed by resampling the kernel initial parameters, from which we observed no dependence of training on the initial parameters. As discussed in line 802, the GP-based models were trained by the ADAM optimizer, with a learning rate of 0.01 for 3000 gradient steps, and other parameters are the default values ​​in pytorch.
>
> > **Question 3**: Why did you mostly focus on performance of the vector field? If the point is learning the dynamics, metrics that look at the dynamics directly would be preferable. Why not prediction of the future state/trajectory? Why prediction only measured through VPT?
>
> **Response**: In our experiments, we evaluate the performance of the models in predicting both vector fields and state trajectories, with the former measured by the root mean squared error (RMSE) and the latter measured by the valid prediction time (VPT). The reason why we chose VPT is that many recent studies in learning dynamical systems [4,5,6] suggested that the RMSEs of state trajectories over long time horizons can be misleading indicators. As an example in the pendulum dataset, a trajectory remaining stationary at its initial position may have lower RMSE compared to a trajectory recovering the oscillatory behavior correctly but having a slight shift in angular velocity. Therefore, VPT has become a popular metric for measuring a model's ability to predict state trajectories.
>
> ---
> Thank you again for the constructive comments. We hope these explanations could address your concerns. Any further questions or suggestions would be greatly appreciated.
>
> ---
> ## References
>
> [1] Liu, Ziming, and Max Tegmark. "Machine learning hidden symmetries." Physical Review Letters 128.18 (2022): 180201.
>
> [2] Desai, Krish, Benjamin Nachman, and Jesse Thaler. "Symmetry discovery with deep learning." Physical Review D 105.9 (2022): 096031.
>
> [3] Bhatia, Harsh, et al. "The Helmholtz-Hodge decomposition—a survey." IEEE Transactions on visualization and computer graphics 19.8 (2012): 1386-1404.
>
> [4] Matsubara, Takashi, and Takaharu Yaguchi. "Finde: Neural differential equations for finding and preserving invariant quantities." arXiv preprint arXiv:2210.00272 (2022).
>
> [5] Jin, Pengzhan, et al. "SympNets: Intrinsic structure-preserving symplectic networks for identifying Hamiltonian systems." Neural Networks 132 (2020): 166-179.
>
> [6] Vlachas, Pantelis-Rafail, et al. "Backpropagation algorithms and reservoir computing in recurrent neural networks for the forecasting of complex spatiotemporal dynamics." Neural Networks 126 (2020): 191-217.

---

> > ### Comment · Reviewer_KheP · 2024-08-10
> >
> > Thank you for your rebuttal. However, I still believe that the paper would benefit from comparisons to relevant baselines (besides GPs and HNNs). Specifically, including a comparison to other dynamical systems reconstruction methods would provide valuable context and help to clarify the advantages of your approach. Such comparisons could significantly strengthen the paper by demonstrating its effectiveness relative to existing techniques.
> >
> > I will keep my score as is.

---

### Official Review · Reviewer_X44d · 2024-07-08

**Soundness:** 4
**Presentation:** 4
**Contribution:** 4
**Rating:** 7
**Confidence:** 4

**Summary:**

The authors tackle the problem of modelling dynamical systems in scenarios where it may not be possible to straightforward to determine the exact form of the ODEs governing the system and optimise their parameters directly. Whilst physics-informed Bayesian models which learn divergence-free vector fields are capable of extrapolating dynamics effectively in an interpretable manner, they cannot represent certain common real-world behaviours such as dissipation. Proposed in this work is a Gaussian process (GP) which combines the utility of the div-free vector field with a curl-free vector field, an approach which is theoretically motivated by the Helmholtz-Hodge decomposition which states that many vector fields may be decomposed into a sum of a div-free and curl-free field. This approach (termed the SPHHD-GP) allows for a much wider range of dynamical systems to be modelled without strict assumptions on the form of the system, and also permits a symmetry preserving representation which allows for enhanced interpretability of the learned dynamics. Experimental results are presented which show the efficacy of the approach on tasks such as learning Hamiltonian dynamics from noisy data, learning chaotic dynamics and modelling ocean currents.

**Strengths:**

-	The approach taken is novel, but not unnecessarily complex. The theoretical insight regarding the Helmholtz-Hodge decomposition is combined with a basic building block of GP-based modelling (additive combination of GPs) in order to yield a technique which consistently outperforms all the relevant baselines presented by the authors.
-	The quality of the work carried out in general is very good; the experiments are thorough and well documented overall. The Hamiltonian dynamics experiments are effectively a standard benchmark in this area, the chaotic system presents a further challenge for the model, and the ocean current experiment is a real-world example of how this approach may be useful in a practical setting. The performance of the SPHHD-GP across all of these experiments is impressive.
-	The clarity of presentation is also very good, and I believe that the work would be accessible both to researchers from a dynamical systems background without in-depth knowledge of GPs, and vice versa. The narrative of the paper is well crafted, from problem setting to initial formulation of the model, onto discussion of limitations regarding identifiability, and finally the implementation of symmetry-based constraints which address these limitations.

**Weaknesses:**

-	From the text it appears the ocean current experiment was performed using a sparse GP implementation, whereas the rest of the experiments were performed using exact inference, is this correct? This is only mentioned very briefly in passing and few details given; I know the details of the sparse GP formulation are probably a given to readers with a background in the area, but I think adding some detail to the appendix on how this was implemented would be useful to many readers.
-	Several of the derivations, proofs and discussions in the appendix are of sufficient relevance that it would be great if they could be included in the main text, however I’m obviously aware that the authors are constrained by the page limit so I wouldn’t expect any alterations based on this. I think the amount of content more just speaks to the fact the work has been carried out in a thorough manner.

**Questions:**

-	Please address the point mentioned in the weaknesses section regarding adding some brief details about the sparse implementation into the appendix and make some reference to this in the main text (maybe where the sparse GP framework is mentioned in Section 6.3).
-	Related to the above point, could you speak to how the sparse formulation performs empirically compared to the exact approach on problems for which both are computationally feasible (i.e., sections 6.1 and 6.2)?

**Limitations:**

The authors address the main limitation of the model in Appendix I which is the considerable computational complexity of the model, $\mathcal{O}(m^3 n^3)$, although this approach can also be implemented using sparse GPs instead to reduce this. The authors also briefly mention in Appendix K how the model has a wide range of applications, but from a societal impact perspective it is important that predictions and uncertainty estimates are rigorously evaluated. As I don’t think the work has significant potential for negative impact, I think this is adequate.

---

> ### Author Rebuttal · Authors · 2024-08-07
>
> # Response to Reviewer X44d
>
> Thank you very much for your thoughtful review and constructive comments. We are very pleased that you recognize the importance and contribution of our work. We have carefully gone through your comments and suggestions, and we believe addressing these points in the manuscript would indeed make the paper better.
>
> ---
>
> > **Question 1**: Please address the point mentioned in the weaknesses section regarding adding some brief details about the sparse implementation into the appendix and make some reference to this in the main text (maybe where the sparse GP framework is mentioned in Section 6.3).
>
> **Response**: Following your suggestion, we will add some brief details about the implementation of the sparse GP into the appendix. The key idea of this sparse GP framework is to introduce pseudo-inputs, which form a smaller set of data serving as a compact representation of the original data. These pseudo-inputs are not necessarily a subset of the training data but are optimized to capture the essential structure of the dataset. In our experiments in Section 6.3, 200 training data points are randomly selected as the initial pseudo-inputs, whose locations are learned jointly with kernel parameters through the ADAM optimizer. These pseudo-inputs greatly reduce the computational complexity while ensuring that the sparse GP model maintains a high prediction accuracy.
>
> > **Question 2**: Related to the above point, could you speak to how the sparse formulation performs empirically compared to the exact approach on problems for which both are computationally feasible (i.e., sections 6.1 and 6.2)?
>
> **Response**: Considering the computational feasibility, the experiments in Sections 6.1 and 6.2 were performed using exact inference. The sparse GP framework can achieve similar prediction accuracy to the exact inference as long as sufficient pseudo-inputs are employed.
>
> ---
> Thank you again for the constructive comments. We hope these explanations could address your concerns. Any further questions or suggestions would be greatly appreciated.

---

> > ### Comment · Reviewer_X44d · 2024-08-08
> > **Rebuttal response**
> >
> > Thank you very much to the authors for taking the time to clarify those points, I appreciate it's primarily details that will be apparent to people closely familiar with GPs, but I do think expanding on them will broaden the accessibility of the paper to those coming more from a physics background.
> >
> > I keep my score unchanged and recommend acceptance.

---

### Official Review · Reviewer_w6Wk · 2024-07-09

**Soundness:** 3
**Presentation:** 3
**Contribution:** 3
**Rating:** 6
**Confidence:** 4

**Summary:**

The paper formulates a vector-valued GP model to infer unknown vector fields. The authors discuss how to impose symmetry-based constraints on the GP models to ensure physically meaningful decompositions. The paper provides theoretical proofs to support the construction of curl-free and divergence-free vector fields that preserve desired symmetries. The model's effectiveness is validated through experiments on various dynamical systems, including dissipative Hamiltonian dynamics and chaotic systems.

**Strengths:**

- The curl-free and divergence-free kernel considering symmetry constraints is novel.
- The manuscript is generally easy to read.
- The model is evaluated on multiple dynamical systems.

**Weaknesses:**

- There are missing citations for relevant existing methods.
- There is no discussion on computational complexity.

**Questions:**

- In [R1], a Gaussian Process for dissipative Hamiltonian systems has been proposed. Can HHD-GP be considered a generalization of [R1]? What are the essential differences between SSGP and HHD-GP? While SPHHD-GP is clearly novel, it was unclear whether HHD-GP is new compared to [R1].
- Please add a discussion on computational complexity. Specifically, numerical integration is used to consider symmetry constraints, but could this be a computational bottleneck?
- In Figure 3, the error of HHD-GP appears to increase with the number of samples, which seems unnatural. Shouldn't this be analyzed in more detail?

[R1] Yusuke Tanaka, Tomoharu Iwata, Naonori Ueda, Symplectic Spectrum Gaussian Processes: Learning Hamiltonians from Noisy and Sparse Data, NeurIPS, 2022.

**Limitations:**

The differences between HHD-GP and existing methods (such as [R1]) should be thoroughly discussed.

---

> ### Author Rebuttal · Authors · 2024-08-07
>
> # Response to Reviewer w6Wk
>
> Thank you very much for your constructive comments and for taking the time to review our manuscript. We've carefully read your comments and our response is as follows.
>
> ---
>
> > **Question 1**: In [R1], a Gaussian Process for dissipative Hamiltonian systems has been proposed. Can HHD-GP be considered a generalization of [R1]? What are the essential differences between SSGP and HHD-GP? While SPHHD-GP is clearly novel, it was unclear whether HHD-GP is new compared to [R1].
>
> **Response**: SSGP and our models (HHD-GP and SPHHD-GP) study the problem of learning dynamical systems from different perspectives and scopes, leading to essential differences in their construction of kernels. Specifically, we emphasize that our kernels are constructed from the perspective of satisfying certain differential invariants (either free of divergence or of curl), rather than from physically governing equations (e.g. the Hamiltonian equation used by SSGP). Therefore, HHD-GP has a wider scope of applicability, which is applicable to any physical system that can be described by a smooth vector field, but SSGP can only be used for dissipative Hamiltonian systems. For example, SSGP cannot be used to model the Chua circuit, a chaotic system used in our experiments, because this system cannot be described by the Hamiltonian equation. Hamiltonian vector fields are a specific type of divergence-free vector fields. From this perspective, HHD-GP can indeed be considered as a generalization of SSGP. Furthermore, although SSGP combines a Hamiltonian kernel with an additive dissipation kernel, it does not consider the non-identifibility problem prevalent in additive models, making the model struggle with learning system components from noisy data. In contrast, we solve the non-identifibility problem in HHD-GP by incorporating symmetries of underlying dynamics. We believe this is an original and valuable contribution to the field of machine learning for dynamical systems. Thank you for providing us with the opportunity to further elaborate on the contributions of our work, which we will include in the future manuscript as you suggest.
>
> > **Question 2**: Please add a discussion on computational complexity. Specifically, numerical integration is used to consider symmetry constraints, but could this be a computational bottleneck?
>
> **Response**: When the symmetry-preserving kernel admits no closed form, numerical integration is exploited to approximate the kernel and compute the covariance, whose computational complexity grows linearly with the number of discrete points. To investigate the impact of the number of discrete points on model performance, we performed experiments on learning with the kernel maintaining rotation symmetry approximated by varying amount of discrete rotations. The results are given in the table below, from which we can observe that the performance of the kernel approximation improves rapidly with increasing number of discrete rotations. Kernels with rotational symmetry were well approximated even with a small number of samples (e.g., n = 8), which ensures that the numerical integration could not be a computational bottleneck. The computational complexity of our method is dominated by the Cholesky decomposition to invert the covariance matrix, which grows cubically with the number of training data points and the input dimension.
>
> | Number of rotation samples | $n=1$ | $n=4$  | $n=8$  | $n=36$ | $n=100$ |
> |-----|-----|-----|-----|-----|-----|
> | RMSE $(\times 10^{-2})$ |  5.76  |   3.12   |  1.96 | 1.74 | 1.67
>
>
> > **Question 3**: In Figure 3, the error of HHD-GP appears to increase with the number of samples, which seems unnatural. Shouldn't this be analyzed in more detail?
>
> **Response**: Figure 3 provides the experimental results for the RMSE of energy prediction as the number of training data increases, which demonstrate the non-identifiability problem suffered by HHD-GP. The covariance matrix equation ($K_* = K_{curl} + K_{div} + \\Sigma$, Eq.6 in the manuscript) shows that when the HHD-GP is used to learn the decomposition, the effects of different components can be treated as observation noise to each other. However, the non-uniqueness of HHD makes the HHD-GP model non-identifiable, making it struggle to distinguish between the different factors contributing to the data. Therefore, more uncertainty in model parameters is introduced with more data, leading to an increase in prediction error as the number of training data increases. This inability of the model parameters to converge to their true values as the amount of data increases indicates a lack of model consistency, which is a necessary condition for a model to be identifiable, as discussed from line 321 of the manuscript. We will add more details to explain the experimental results, following your suggestion.
>
> ---
> Thank you again for the constructive comments. We hope these explanations could address your concerns. Any further questions or suggestions would be greatly appreciated.

---

> ### Comment · Reviewer_w6Wk · 2024-08-12
>
> Thank you for your reply. My concerns are now largely resolved, and the differences between the proposed model and SSGP are much clearer. I will raise my score from 5 to 6.

---

### Official Review · Reviewer_DJX1 · 2024-07-15

**Soundness:** 3
**Presentation:** 3
**Contribution:** 3
**Rating:** 7
**Confidence:** 3

**Summary:**

This paper derives a G-equivariant versions of the both curl (through the Haar Integration kernel) and divergence free (through GIM kernel) kernels. This is then used to define a prior over the Helmholtz decomposition and is identifiable wrt to relevant functions in the Euclidean group (translation, rotation, reflection etc). This is demonstrated on a range of examples.

**Strengths:**

This is a well written, clear paper, that improves on recent work with convincing experiments.

**Weaknesses:**

1.	The justification for how and why the curl and div free kernels should be invariant is not given until the experiments/the appendix. It would be nice if this was discussed in section 5.
2.	The fact that the actual symmetries required is problem specific is not really clear in the main paper. In the appendix it is stated, but it would be good if this was made more explicit in the main paper.

**Questions:**

1.	Is the proposed symmetry-preserving Helmholtz decomposition as expressive as non-symmetry preserving one? Ie any (sufficient smooth) field can be decomposed into curl and divergence free part. Is the same for the symmetric-preserving decomposition?
2.	On line 328 it is stated that mirror symmetry is incorporated into the div-free kernel. Where is this kernel given?
3.	Are there general symmetries that should be held or is this always going to be problem/application specific?
4.	When the kernels do not admit a closed form how sensitive are they to approximations? And does this significantly affect the computational cost?
5.	Is there any impact of using sparse GPs? Ie are the sparse models still guaranteed to be G-invariant?
6.	Relevant GP citations:
    o	Equivariant Learning of Stochastic Fields: Gaussian Processes and Steerable Conditional Neural Processes, Holderrieth et al
    o	Vector-valued Gaussian Processes on Riemannian Manifolds via Gauge Equivariant Projected Kernels – Hutchinson et al

**Limitations:**

yes

---

> ### Author Rebuttal · Authors · 2024-08-07
>
> # Response to Reviewer DJX1
>
> We are very grateful for your valuable comments and acknowledgement of our main contributions. We greatly appreciate the time and effort you put into reviewing our paper. Below are our responses to your questions.
>
> ---
>
> > **Question 1**: Is the proposed symmetry-preserving Helmholtz decomposition as expressive as non-symmetry preserving one? Ie any (sufficient smooth) field can be decomposed into curl and divergence free part. Is the same for the symmetric-preserving decomposition?
>
> **Response**: The the Helmholtz-Hodge decomposition involves breaking down a vector field into its divergence-free and curl-free parts, which may not necessarily preserve the symmetries of the original vector field. This is because the decomposition process is based on mathematical properties (curl and divergence) rather than on the symmetries of the field. However, the symmetry-preserving Helmholtz-Hodge decomposition always exists for physical systems due to the physical relevance of their divergence-free and curl-free components. These physical relevance aligns well with the symmetries of underlying systems. The Helmholtz-Hodge decomposition respects these physical relevance, which naturally leads to the preservation of symmetries.
>
> > **Question 2**: On line 328 it is stated that mirror symmetry is incorporated into the div-free kernel. Where is this kernel given?
>
> **Response**: Do you mean the div-free kernel mentioned in line 378? We apologize that we did not provide details about this div-free kernel in the manuscript. To make our GP model identifiable, we constructed the div-free kernel by incorporating mirror symmetry with respect to the square boundary $[0, 1] \\times [0, 1]$. The symmetry group is given by
> $$
> G = \\left \\{
> \\begin{bmatrix} 1 & 0 & 0\\\\ 0& 1 & 0\\\\ 0 & 0 & 1 \\end{bmatrix},
> \\begin{bmatrix} 1 & 0 & 0\\\\ 0& -1 & 0\\\\ 0 & 0 & 1 \\end{bmatrix},
> \\begin{bmatrix} -1 & 0 & 2\\\\ 0& -1 & 0\\\\ 0 & 0 & 1 \\end{bmatrix},
> \\begin{bmatrix} -1 & 0 & 2\\\\ 0& 1 & 0\\\\ 0 & 0 & 1 \\end{bmatrix},
> \\begin{bmatrix} -1 & 0 & 2\\\\ 0& -1 & 2\\\\ 0 & 0 & 1 \\end{bmatrix},
> \\begin{bmatrix} 1 & 0 & 0\\\\ 0& -1 & 2\\\\ 0 & 0 & 1 \\end{bmatrix},
> \\begin{bmatrix} -1 & 0 & 0\\\\ 0& -1 & 2\\\\ 0 & 0 & 1 \\end{bmatrix},
> \\begin{bmatrix} -1 & 0 & 0\\\\ 0& 1 & 0\\\\ 0 & 0 & 1 \\end{bmatrix},
> \\begin{bmatrix} -1 & 0 & 0\\\\ 0& -1 & 0\\\\ 0 & 0 & 1 \\end{bmatrix}
> \\right \\}，
> $$
> where the group elements are the mirror transformation matrices applied to homogeneous coordinates $(x, y, 1)$. Theorem 5.3 in our manuscript shows that a $G$-equivariant div-free vector field can be constructed from a vector potential with the same symmetry. So, we constructed the kernel of the vector potential by using Eq.17 to integrate a *squared exponential kernel* over the symmetry group above. This potential kernel was then substituted into Eq.18 to construct the div-free kernel. The partial derivatives involve in Eq.18 were calculated by automatic differentiation in PyTorch, so we did not need to derive the analytic expression of the div-free kernel. We will add these details in the future manuscript. Thank you very much for pointing out this issue.
>
>
> > **Question 3**: Are there general symmetries that should be held or is this always going to be problem/application specific?
>
> **Response**: There is no general symmetry that must be maintained. Although we have indicated in the appendix that the actual required symmetry is problem-specific, we will make this more explicit in the main paper based on your suggestion.
>
> > **Question 4**: When the kernels do not admit a closed form, how sensitive are they to approximations? And does this significantly affect the computational cost?
>
> **Response**: The kernel maintaining rotation symmetry is constructed by integrating over the continuous rotation group, and this kernel has no closed form (Eq.72). So we numerically approximated this kernel by sampling discrete rotations. To investigate the impact of this kernel approximation, we added experiments on learning with varying number of sampling rotations. The results are given in the table below, from which we can observe that the performance of the kernel approximation improves rapidly with increasing number of sampling rotations. Kernels with rotational symmetry were well approximated even with a small number of samples (e.g., n = 8), which ensures that approximating the kernel by numerical integration does not significantly improve the computational cost. (The computational complexity increases linearly with the number of discrete rotation samples.)
>
> | Number of rotation samples | $n=1$ | $n=4$  | $n=8$  | $n=36$ | $n=100$ |
> |-----|-----|-----|-----|-----|-----|
> | RMSE $(\\times 10^{-2})$ |  5.76  |   3.12   |  1.96 | 1.74 | 1.67
>
> > **Question 5**: Is there any impact of using sparse GPs? Ie are the sparse models still guaranteed to be G-invariant?
>
> **Response**: In our method, the symmetry constraints are imposed to GP models by disigning suitable kernels. The sparse GP model we used reduces the computational complexity by using a smaller set of inducing points to compactly represent the original data, which do not alter the kernel's structure and properties. Therefore, the symmetries can still be respected in the sparse GP model.
>
> > **Question 6**: Relevant GP citations: o Equivariant Learning of Stochastic Fields: Gaussian Processes and Steerable Conditional Neural Processes, Holderrieth et al o Vector-valued Gaussian Processes on Riemannian Manifolds via Gauge Equivariant Projected Kernels – Hutchinson et al
>
> **Response**: Thank you for providing these two works, which will be cited in our future manuscript.
>
> ---
> Thank you again for the constructive comments. We hope these explanations could address your concerns. Any further questions or suggestions would be greatly appreciated.

---

> > ### Comment · Reviewer_DJX1 · 2024-08-12
> >
> > Thank you for your thorough response. I think including the discussions above in the paper will improve an already well-written paper.  I have no more questions and maintain my original score of acceptance.

---

### Decision · Program_Chairs · 2024-09-25

**Decision:**

Accept (poster)

**Comment:**

Reviewers agree that this manuscript on gaussian processes kernels for the Helmholtz-Hodge Decomposition of dynamical systems is clearly written and novel. This is an exciting contribution with many potential uses.